# Coordinated neuronal ensembles in primary auditory cortical columns

Jermyn Z See[1,2,3,4], Craig A Atencio[1,2,3], Vikaas S Sohal[1,4], Christoph E Schreiner[1,2,3]*

[1]UCSF Center for Integrative Neuroscience, University of California, San Francisco, San Francisco, United States; [2]Coleman Memorial Laboratory, University of California, San Francisco, San Francisco, United States; [3]Department of Otolaryngology – Head and Neck Surgery, University of California, San Francisco, San Francisco, United States; [4]Department of Psychiatry, University of California, San Francisco, United States

**Abstract** The synchronous activity of groups of neurons is increasingly thought to be important in cortical information processing and transmission. However, most studies of processing in the primary auditory cortex (AI) have viewed neurons as independent filters; little is known about how coordinated AI neuronal activity is expressed throughout cortical columns and how it might enhance the processing of auditory information. To address this, we recorded from populations of neurons in AI cortical columns of anesthetized rats and, using dimensionality reduction techniques, identified multiple coordinated neuronal ensembles (cNEs), which are groups of neurons with reliable synchronous activity. We show that cNEs reflect local network configurations with enhanced information encoding properties that cannot be accounted for by stimulus-driven synchronization alone. Furthermore, similar cNEs were identified in both spontaneous and evoked activity, indicating that columnar cNEs are stable functional constructs that may represent principal units of information processing in AI.

DOI: https://doi.org/10.7554/eLife.35587.001

*For correspondence:
chris@phy.ucsf.edu

**Competing interests:** The authors declare that no competing interests exist.

## Introduction

How individual neurons work together to encode sensory information and influence behavior remains one of the fundamental questions in systems neuroscience. Single neurons have been traditionally considered to be the basic functional unit of the brain and much of our understanding of how the brain encodes sensory stimuli or motor output comes from decades of single-unit studies. Single-unit activity in isolation, however, is often insufficient to account for observed sensory or motor behaviors (*Bizley et al., 2010*; *Britten et al., 1996*; *Engineer et al., 2008*; *Georgopoulos et al., 1986*; *Herzfeld et al., 2015*; *Paninski et al., 2004*).

Technological advances in large-scale recordings, including calcium imaging and high-density multi-channel electrodes, have allowed the monitoring of the simultaneous activity of large populations of neurons. This has led to the demonstration of coordinated activity within groups of recorded neurons, identified and verified by statistical approaches (*Billeh et al., 2014*; *Gourévitch and Eggermont, 2010*; *Lopes-dos-Santos et al., 2013*; *Miller et al., 2014*; *Peyrache et al., 2010*; *Pipa et al., 2008*).

These concerted neuronal activities are postulated to be local network events that reflect improved associations with decision-making, predictions of perceptual events, memory formation, and behavioral performance over isolated single-unit activity (*Bathellier et al., 2012*; *Bell et al., 2016*; *Gulati et al., 2014*; *Ince et al., 2013*; *Kiani et al., 2014*; *Laubach et al., 2000*; *Peyrache et al., 2009*; *Reimann et al., 2017*). Coordinated ensembles have also been postulated to

be elementary units of information processing in the brain (for review see *Buzsáki, 2010*; *Harris and Mrsic-Flogel, 2013*; *Yuste, 2015*). However, basic statistical properties of these ensembles, such as cellular composition, extent, stability, and functional roles, including selectivity of information extraction and reliability of transmission, are not yet well understood.

This is especially true in the auditory cortex (AC). Most of what we understand about AC function is based on single-unit and general population analyses. Single-unit studies in the AC have focused on characterizing receptive fields, treating AC neurons as arrays of (nearly) linear filters (*Aertsen and Johannesma, 1981*; *Atencio et al., 2012*; *Calabrese et al., 2011*; *Thorson et al., 2015*). Population activity in the AC, often based on indiscriminate pooling of single- or multiple-unit activity, has been shown to correlate with an animal's perception of simple sounds (*Bathellier et al., 2012*; *Rodgers and DeWeese, 2014*) and can be used to decode aspects of acoustic information (*Miller and Recanzone, 2009*; *Brasselet et al., 2012*; *Ince et al., 2013*; *David and Shamma, 2013*; *Abrams et al., 2017*). However, these studies did not identify groups of cooperating neurons, treating all simultaneously recorded neurons as equivalent information-processing entities. Meanwhile, synchronous activity between local pairs of AC neurons can reflect shared and unique stimulus aspects, enabling enhanced information encoding (*Atencio and Schreiner, 2016*; *Atencio and Schreiner, 2013*; *Brosch and Schreiner, 1999*; *Eggermont, 2006*; *1992*; *2013*; *Gourévitch and Eggermont, 2010*). While neuronal pairs represent the smallest possible ensemble, larger columnar or distributed networks of synchronously active neurons could reveal specific and particularly meaningful information-bearing network events (*Bathellier et al., 2012*; *Bharmauria et al., 2016*). One particular issue is whether coordinated ensemble events can be meaningfully distinguished from coincidental events that result from stimulus synchrony and overlaps in the receptive fields of neurons. Therefore, to better understand network encoding, the membership criteria, structure and function of synchronously firing neuronal groups must be examined to distinguish between coordinated and coincidental neuronal ensembles.

Here, we identified and characterized coordinated neuronal ensembles (cNEs) from the synchronous activity across populations of neurons within auditory cortical columns. We performed high-density extracellular electrophysiological recordings across several laminae within primary auditory cortical (AI) columns of rats using broadband dynamic noises (*Atencio and Schreiner, 2010a*; *2010b*). The cortical column is a fundamental organizing substrate within the AC, and relates neurons with similar preferred frequencies (*Wallace and Palmer, 2008* (gerbil); *Atencio and Schreiner, 2010a*, *2010b* (cat); *Guo et al., 2012* (mouse); but see *Winkowski and Kanold, 2013*). Recording from the column therefore increased the likelihood of encountering stimulus-driven, temporally aligned spiking activity. We chose the ketamine-anesthetized preparation to enhance the stability of the recordings over time and to reduce the influence of state changes on neural networks; the awake (*Fritz et al., 2005*; *McGinley et al., 2015*; *Okun et al., 2010*; *Poulet and Petersen, 2008*) or urethane-anesthetized (*Clement et al., 2008*; *Marguet and Harris, 2011*; *Pagliardini et al., 2013*) rat cortex would undergo state changes more frequently, potentially affecting the composition of cNEs. We applied dimensionality reduction techniques to recorded populations and identified cNEs, which consisted of neurons that exhibited sharply synchronous activity (*Lopes-dos-Santos et al., 2013*). Our results show that cNEs can be accurately and reliably detected, have higher coincident firing than can be explained from receptive field similarity or second-order (pairwise) correlations, and have active events that occur more frequently than expected from population recordings. cNEs identified from spontaneous and evoked epochs were highly similar, implying that cNEs represent functional networks that are not dependent on stimulus presentation. Our results also show that cNEs encode auditory information more reliably than single neurons or random groups of neurons with similar receptive fields and support the notion that cNEs represent unique and functionally significant units of processing in AI.

## Results

### Heterogeneous functional connectivity between columnar AI neurons

We used high-density recording electrodes (*Figure 1A*) to simultaneously record the activity of populations of neurons across supragranular and granular laminae within AI columns. Presenting dynamic, broadband stimuli with varying statistics (dynamic moving ripple (DMR) or ripple noise

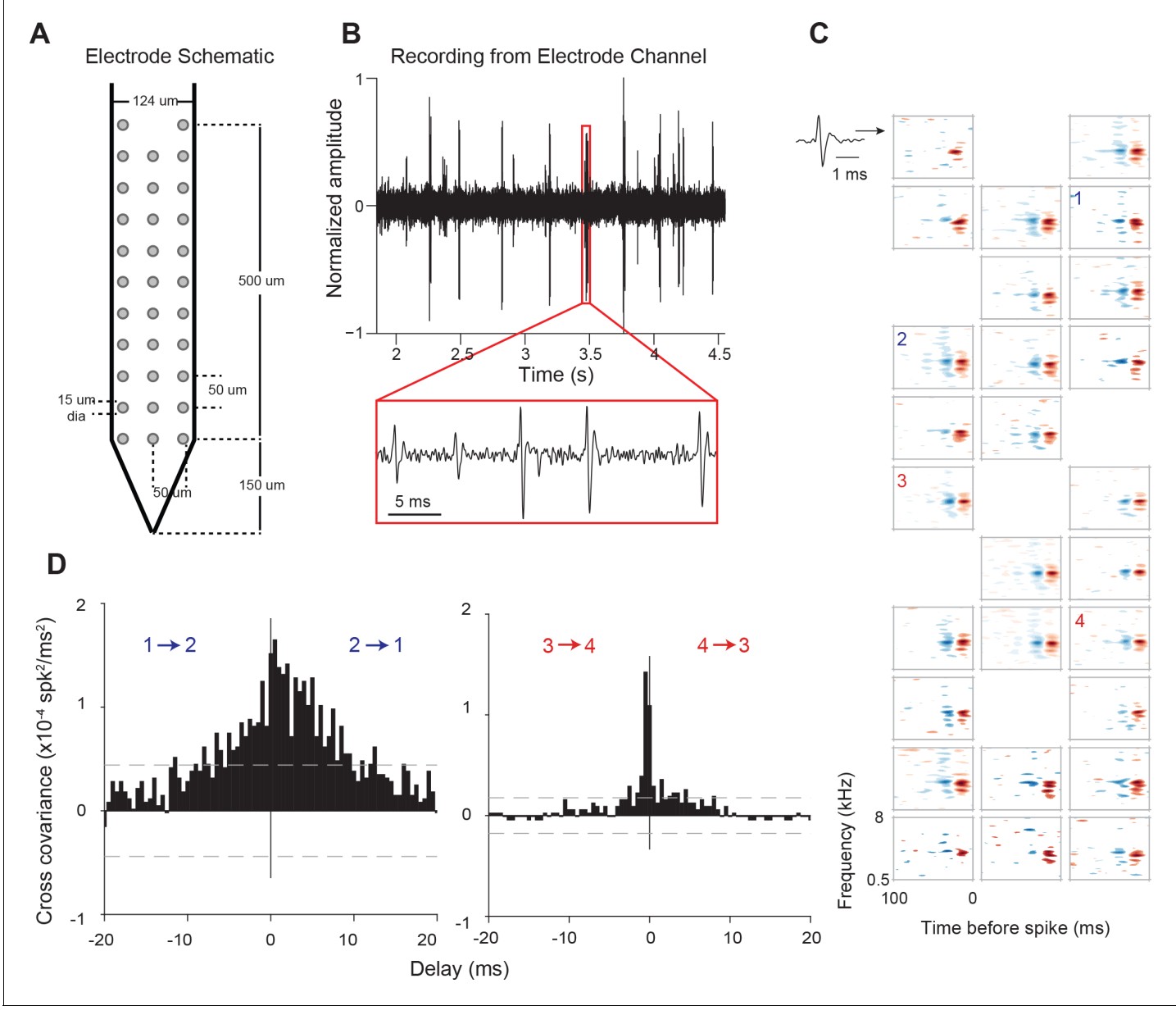

**Figure 1.** Electrode used, sample recording, STRFs, and PWCs. (A) Electrode schematic. (B) Sample recording from one electrode channel. (C) Sample STRFs obtained after spike sorting. Numbers 1–4 indicate the positions and STRFs of pairs of neurons whose PWCs are plotted in (D). (D) Example PWCs from two pairs of neurons (1–2 and 3–4). Neurons 3 and 4 exhibit a sharper PWC than neurons 1 and 2 despite having approximately similar STRFs and pairwise distances. Dashed lines represent 99% confidence intervals.

DOI: https://doi.org/10.7554/eLife.35587.002

(RN); see Materials and methods) allowed us to estimate the spectro-temporal receptive fields (STRFs) of identified neurons (*Figure 1C*). STRFs of neurons from the same columnar penetration showed general similarities in characteristic frequency and frequency tuning, and displayed minor variations in spectral and temporal details (see example in *Figure 1C*), similar to cat cortical columnar recordings (*Atencio and Schreiner, 2010a*).

We first examined response synchrony between simultaneously recorded single units using Pearson's correlation analysis at 10-ms temporal resolution. Pairs of neurons in the same column with similar STRFs could nonetheless exhibit widely differing pairwise correlation (PWC) functions (*Figure 1D*). For example, the PWC between pairs of neurons with similar receptive fields, recorded

from channels of equal intra-pair distances, could differ in peak correlation magnitude, width and delay (*Figure 1C and D*). Pearson's correlation values (or correlation coefficients), PWC absolute peak delays and sharpness are depicted in *Figure 2A* for one neuron (#15) with all other recorded neurons in the same penetration. Significant PWCs were evident for most pairs in this penetration but low correlation coefficients and broader correlation widths were dominant. Across all

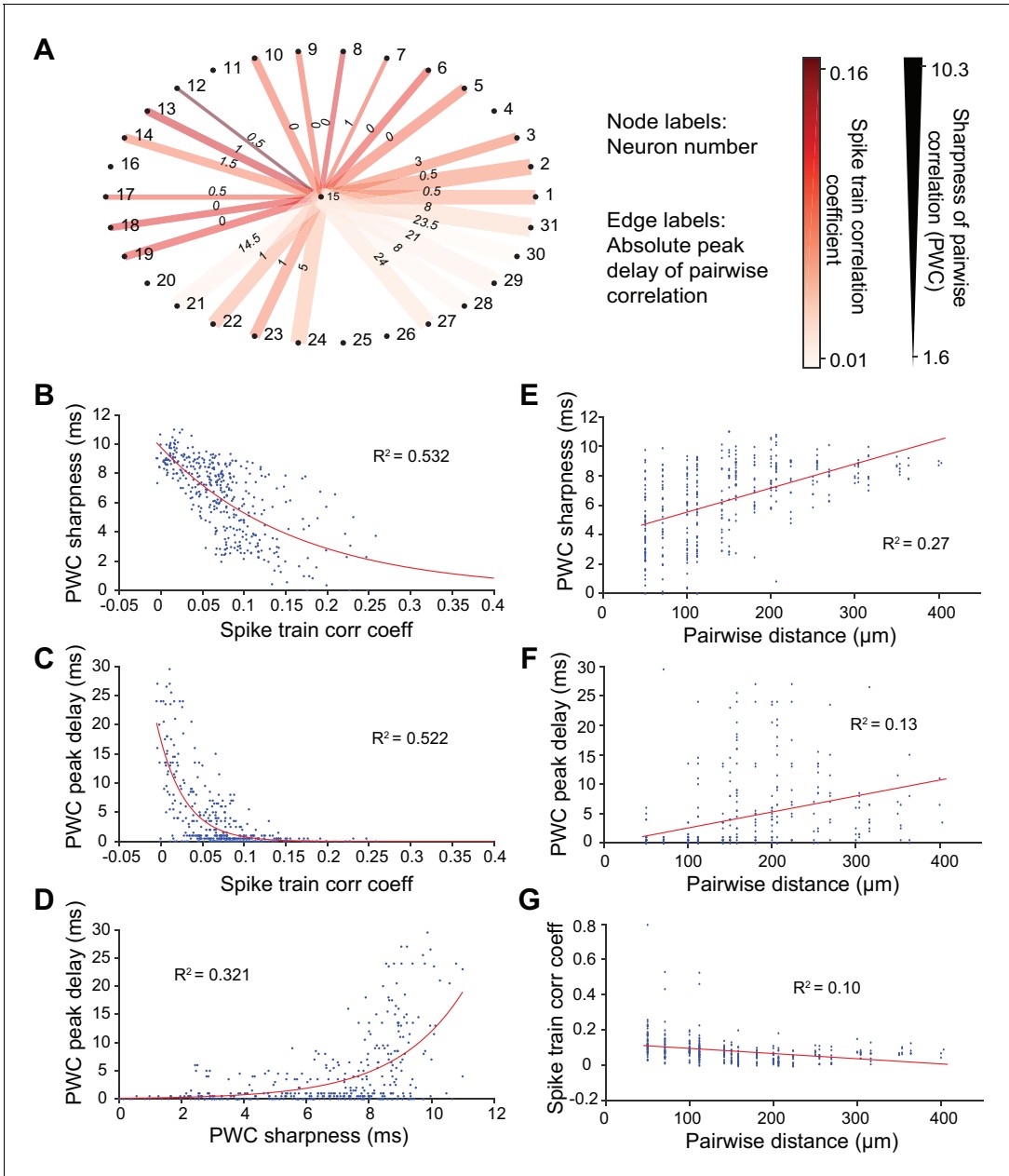

**Figure 2.** Methods for calculating neuronal correlation. (**A**) An example neuron's (#15) correlation with other neurons in a dataset of 31 neurons measured in three different ways. The node labels (outer circle numbers) represent the neuron numbers. The edge labels (inner circle numbers) represent the absolute peak delay of the pairwise correlations (PWCs). The color of the edges represents the spike train correlation coefficient (Pearson's correlation) between pairs of neurons. The thickness of the edges represents the sharpness of the PWCs (see Figure 6G–I for an illustration of how this is computed). (**B**) Sharpness of PWCs against spike train correlation coefficient. (**C**) Peak delay of PWCs against spike train correlation coefficient. (**D**) Peak delay against sharpness of PWCs. All measures of correlation are highly correlated with one another. (**E–G**) Sharpness of PWCs (**E**), peak delay of PWCs (**F**) and spike train correlation coefficient between pairs of neurons (**G**) against pairwise distance. Pairwise distance explains only a small fraction of the variance seen in the different measures of correlation between pairs of neurons.
DOI: https://doi.org/10.7554/eLife.35587.003

penetrations, PWCs were fairly heterogeneous but sharpness and delay covaried with correlation coefficients, accounting for ~50% of the variance (*Figure 2B–2D*).

The degree of synchrony between pairs of neurons often weakens as the distance between neurons within a column increases (*Atencio and Schreiner, 2013*; *Gururangan et al., 2014*). We observed similar trends in columnar recordings, although the correlation between these synchrony aspects and pairwise distance only explained ~10–30% of the total variance ($R^2$ values, *Figure 2E–2G*). Notwithstanding weak spike train correlations, neuronal coordination and synchronization have been widely postulated to play a vital role in information processing and the organization of the nervous system within the cortical column (*Atencio and Schreiner, 2010a*; *Panzeri et al., 2003*). We therefore sought to identify larger columnar groups of neurons that reliably share synchronous firing events, resulting in higher correlated rates of firing than could be accounted for by inter-neuronal distances and shared receptive field properties.

## Identification of coordinated neuronal ensembles (cNEs)

To identify synchronously firing groups of neurons, or coordinated neuronal ensembles (cNEs), we combined two dimensionality reduction techniques - principal component analysis (PCA) and independent component analysis (ICA), following the approach by *Lopes-dos-Santos et al., 2013*. This analysis identifies groups of neurons that repeatedly fire synchronous action potentials throughout the duration of the recording. This approach provides a parametric analysis that identifies both the number of cNEs and the neurons that are members of each cNE. Further, simulations show that this approach is applicable to both large- and small-scale recordings, and that it can be applied to spike trains with Poisson characteristics (*Lopes-dos-Santos et al., 2013*; *Peyrache et al., 2010*). Application of PCA to the simultaneously recorded spike trains determines the number of cNEs above a significance criterion (see Materials and methods). ICA is then applied to the significant eigenvectors or principal components (PCs) to extract the linear combination of the individual neurons' activity associated with each cNE (*Lopes-dos-Santos et al., 2013*). To illustrate this approach, we simulated the spike trains of 8 neurons, where two groups of five neurons had highly synchronous firing, with two neurons being shared between these two groups (*Figure 3A and B*). Each spike train was first binned and z-scored to ensure that cNE identification was not skewed towards neurons with higher firing rates. The correlation coefficients between the spike trains were calculated and represented as a correlation matrix (*Figure 3B*). PCA was applied to the correlation matrix, resulting in a distribution of eigenvalues and corresponding PCs. Each eigenvalue represents the relative importance of each PC. To determine which PCs represented groups of neurons that significantly fired together, we compared the distribution of eigenvalues from the data to the distribution that would be expected for groups of independently firing neurons (*Marčenko and Pastur, 1967*). In the simulated example, there were two significant eigenvalues reflecting the fact that there were two cNEs (*Figure 3C*). However, because neurons 4 and 5 were shared between both cNE #1 and #2, the PCs did not accurately represent the true membership of the two cNEs due to PCA's variance maximization framework (*Figure 3D*). To resolve this ambiguity, we applied ICA to the two most significant PCs, which resulted in two independent components (ICs) that accurately identified the two cNEs (*Figure 3F*). Both the PCs and ICs can be used to recreate the activity of the cNEs over time by projecting them back onto the spike matrix. However, since PCA did not separate the cNEs properly (*Figure 3E*), only the ICA-derived activity correctly reflected the spiking activity of each cNE (*Figure 3G*).

Applying this cNE detection algorithm to an actual AI columnar recording with 38 isolated neurons (*Figure 4D*), we identified nine separate cNEs (*Figure 4B*) with significantly contributing members ranging from 2 to 6 neurons. Each column in *Figure 4B* shows the color-coded IC weight for all neurons in each cNE. Neuronal membership in a cNE was assigned by exceeding a statistical criterion of the IC weight for each neuron in the penetration. Green bars in the weight plot (*Figure 4B*) indicate neurons with significant IC weights. To determine the threshold criterion, we circularly shuffled spike trains within a spike matrix independently over 100 different iterations, ran PCA and ICA on these shuffled spike matrices, and set the threshold at which a neuron belongs to a cNE as 1.5 standard deviations above and below the mean of the distribution of the resulting IC values (*Figure 4C*, see Materials and methods). We used a similar method to determine the minimum magnitude of cNE activity during the recording period that counts as a significant cNE event (*Figure 4E*, see Materials and methods). For example, the cNE analysis revealed that almost every time cNE #2

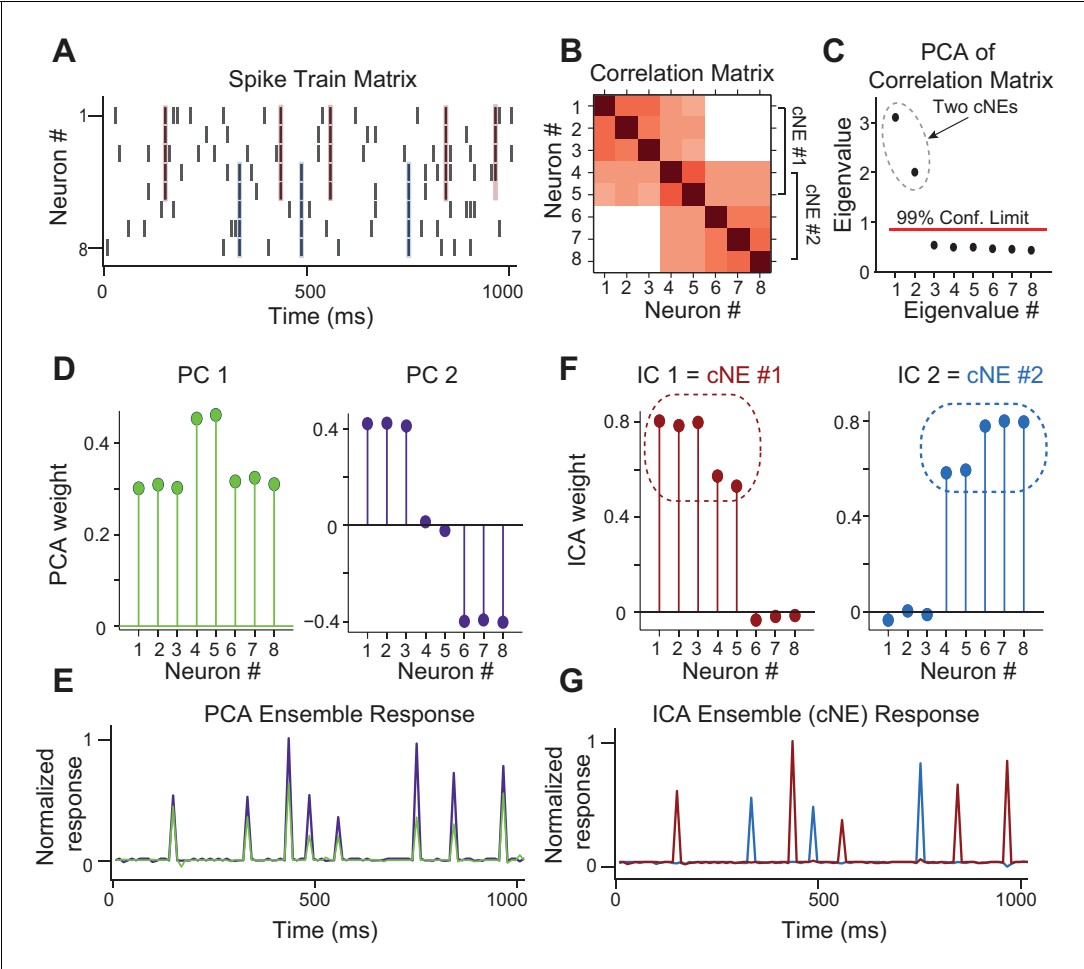

**Figure 3.** Simulated data illustrating the coordinated neuronal ensemble (cNE) detection algorithm. (A) Raster of the simulated data. cNE #1 consists of neurons 1 to 5 while cNE #2 consists of neurons 5 to 8. Neurons 4 and 5 belong to both cNEs. Coincident spiking events between neurons in cNE #1 and cNE #2 are highlighted in red and blue respectively. (B) Correlation matrix of spike train matrix from (A). (C) Eigenvalues from applying PCA to the correlation matrix. The top two eigenvalues are significant and represent the number of cNEs. (D) The eigenvectors (or principal components, PCs) corresponding to the two significant eigenvalues from (C). The PCs do not represent the two cNEs denoted in (A) and (B). (E) Ensemble responses calculated from the projection of the PCs onto the spike train matrix. The PCA ensembles do not separate the activities of cNE #1 and #2 shown in (A). (F) Independent components (ICs) obtained after applying ICA to the two significant eigenvectors from (C). The ICs accurately represent the two cNEs denoted in (A) and (B). (G) Ensemble responses calculated from the projection of the ICs onto the spike train matrix. The ICA ensembles successfully resolve the activities of cNE #1 and #2 shown in (A).

DOI: https://doi.org/10.7554/eLife.35587.004

was deemed active, at least three members (out of a total of 6) of that cNE had to be active (*Figure 4D and E*).

The cNE detection algorithm typically identified several independent cNEs, each capturing moments of joint activity between several of its members. In the case of the example recording, 9 cNEs were extracted. Since cNEs were extracted via ICA, the activities of the different cNEs are necessarily independent of one another across the whole recording period. Yet, as can be seen in the bottom panel of *Figure 4E*, cNEs can fire together with other cNEs (e.g., cNE #2 (red trace) and cNE #3 (green trace) at the 1.1 s and 1.75 s time marks). The cNEs identified in 15 additional columnar recordings are shown in *Figure 4—figure supplement 1*.

We next tested if the identified cNEs were stable and reliable estimates of coordinated group activity that persisted throughout the entire recording. First, we wanted to determine if the cNEs identified from one half of a recording were similar to those identified independently from the other half. We split the recordings into 10 equal parts and applied the cNE algorithm to both sets of

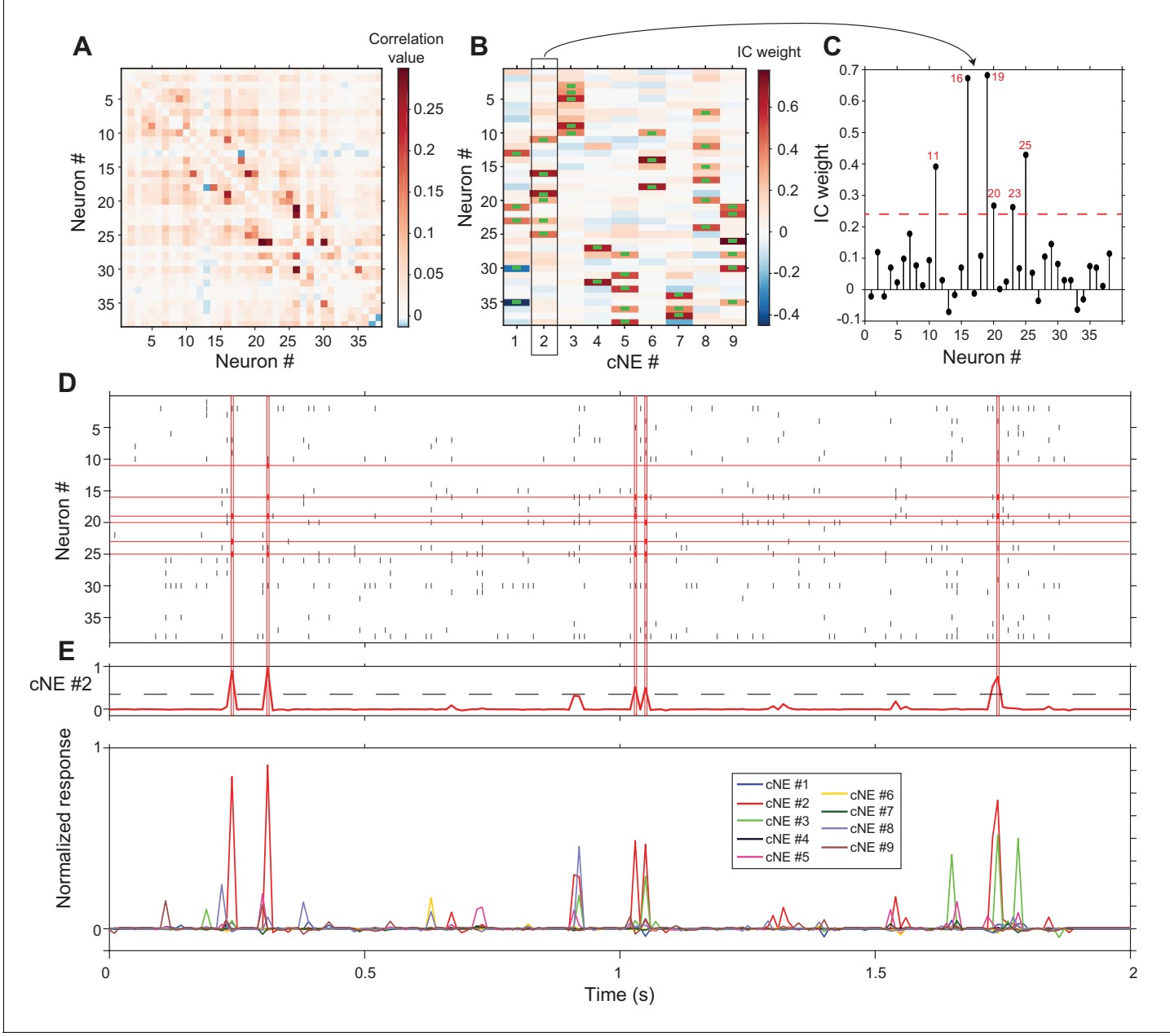

**Figure 4.** cNE detection algorithm for AI neurons. (**A**) Correlation matrix of spike train matrix from (**D**). The diagonal has been set to 0. (**B**) IC weight for each neuron in each cNE. Each column represents one cNE; each row represents one neuron. The IC weight represents the contribution of each neuron to each cNE. The green bars represent neurons that are members of a cNE. (**C**) IC 2 (cNE #2). The red dashed line is the threshold for cNE membership determined by Monte-Carlo methods. The red numbers are the numbers of neurons that are members of the cNE. (**D**) Spike raster of 2 s of real data. Spikes that contribute to instances of cNE #2 activity in (**E**) are in red. (**E**) (top) Trace of cNE #2 activity. The black dashed line represents the threshold for cNE #2 activity estimated via Monte Carlo methods. Peaks that cross the threshold correspond to the coincident neuronal spikes highlighted in red in (**D**). (bottom) Activity traces of the 9 cNEs recovered from this dataset.

DOI: https://doi.org/10.7554/eLife.35587.005

The following figure supplements are available for figure 4:

**Figure supplement 1.** IC weights for each neuron in each cNE for all datasets used (other than the one shown in *Figure 4*).
DOI: https://doi.org/10.7554/eLife.35587.006

**Figure supplement 2.** Detected cNEs are stable throughout the recording.
DOI: https://doi.org/10.7554/eLife.35587.007

**Figure supplement 3.** Detected cNEs are stable throughout the recording.
DOI: https://doi.org/10.7554/eLife.35587.008

interleaved parts of the recording (*Figure 4—figure supplement 2A*). The halves were interleaved to compensate for potential recording instabilities. The correlation coefficients for each of the two sets of independent components were compared (*Figure 4—figure supplement 2B*). To calculate the significance of all correlations within each penetration, spike trains were circularly shuffled, and processed with the cNE detection algorithm (similar to the method to determine the threshold criterion for neuronal membership in cNEs). The resulting ICs were correlated to give a null distribution, and the 99th percentile for each null distribution was set as the significance threshold (*Figure 4—figure supplement 2C*). Approximately 96% of independent components had significant matches (*Figure 4—figure supplement 2D and E*), indicating that the two interleaved parts of the recording contained the same independent components and that the cNEs were present throughout the entire recording.

Secondly, we wanted to test if cNEs detected in a subset of the recorded activity would be able to accurately estimate the cNE activity of the rest of the recording. In this case, we split the recording into two datasets for training (first half) and for testing (second half). Applying the PCA/ICA approach to each dataset, we obtained two corresponding sets of independent components, and projected both sets onto the test data to extract training and test cNE activities (*Figure 4—figure supplement 3A*). We then correlated the training and test cNE activities (*Figure 4—figure supplement 3B*) and took the maximum correlation value for each training cNE as 'matched correlations' (*Figure 4—figure supplement 3D*). The remaining correlation values for all penetrations were treated as 'non-matched correlations' and formed the null distribution, and the 95th percentile of that null distribution was the significance threshold (*Figure 4—figure supplement 3C*). Across all penetrations, approximately 83% of training cNE activity had significant matches to test cNE activity. Thus, cNEs obtained from the first half of the recordings were sufficient to predict the activity of cNEs in the latter half of the recording. Together, these tests show that derived cNEs are stable and reliable descriptors of coordinated neuronal activity.

## cNE identity is dependent on time bin size

In previous studies, the time interval for defining synchronous firing varied widely, from a few milliseconds to several hundred milliseconds (see *Figure 5—source data 1* for a selection of example studies). Reasons for the different choices of time ranges were related to the temporal resolution permitted by the applied methods and the physical distance between considered neuronal elements. We chose a temporal resolution consistent with the time needed for auditory information to traverse the interlaminar microcircuit in AI columns. Using the propagation speed found in the cat and rat, the time required to traverse a rat AI column of approximately 1500 μm in extent is ~10 ms (*Atencio and Schreiner, 2010a*; *Kubota et al., 1997*). The upper limit was defined by the need to avoid stimulus-based temporal properties; the dynamic moving ripple stimulus contained temporal envelope fluctuations up to 40 Hz, resulting in a minimum temporal resolution of 25 ms. We therefore selected 10 ms as a reasonable window of within-column synchrony. Inclusion of synchrony with extracolumnar sources and targets may require analysis with a different window size.

To determine the functional significance of the chosen time window, we binned spikes into varying time bin sizes, obtained cNE activity for each bin size, and then estimated the receptive field information of cNEs detected based on the bin sizes (see Materials and methods). Receptive field information peaked at bin sizes of approximately 10–15 ms (*Figure 5A*), implying that the temporal resolution used to obtain cNEs provided the most information about the receptive field preferences of groups of synchronously firing neurons in AI, further supporting our choice. This temporal resolution also agrees with other studies on the optimal window for dendritic integration (*Branco and Häusser, 2011*; *Polsky et al., 2004*; *Williams and Stuart, 2002*) and spike-timing dependent plasticity (*Bi and Poo, 1998*; *Celikel et al., 2004*; *Markram et al., 1997*).

The size of the time window also affects the number and size of identified cNEs. We would expect that as temporal resolution decreases and the size of the window that defines synchrony increases, more concurrently active neuronal responses are included in each cNE. cNEs that might have been deemed separate in 10-ms time bins might instead be classified as a single cNE when the data is processed using longer time bins, leading to fewer but larger cNEs. Indeed, we found that as the spike train temporal bin size increased, the number of recovered cNEs decreased (*Figure 5B*) and the number of member neurons in each cNE increased (*Figure 5C*).

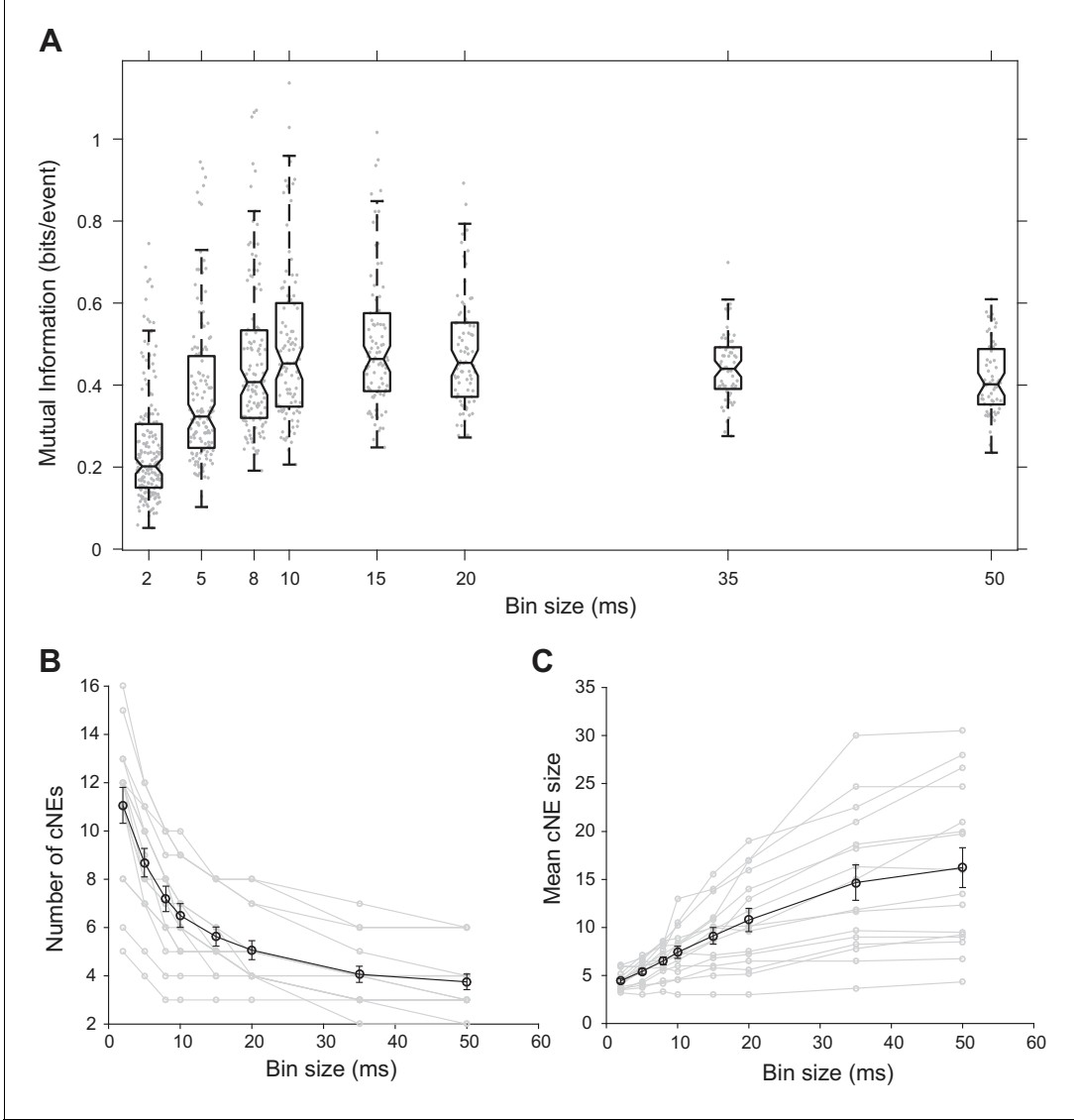

**Figure 5.** Effect of bin size on cNE properties. (**A**) Mutual information between cNE STRFs and the stimulus for different bin sizes. The highest mutual information occurred for bin sizes of approximately 10 to 15 ms. (**B**) Number of detected cNEs decreases with increasing bin size. (**C**) Mean cNE size increases with increasing bin size.

DOI: https://doi.org/10.7554/eLife.35587.009

The following source data is available for figure 5:

**Source data 1.** Example studies that define neuronal synchrony using different methods in different brain areas.

DOI: https://doi.org/10.7554/eLife.35587.010

## Synchrony between cNE members cannot be fully explained by receptive field overlap

Columnar neurons show significant heterogeneity in their PWCs (*Figures 1D* and *2*). First, we sought to confirm that spike train PWCs differ depending on whether pairs of neurons belong to the same cNE. This is to be expected given that cNE membership was based on Pearson's correlation coefficients, which are highly correlated with PWC sharpness (*Figure 2B*). Secondly, we wanted to determine whether the degree of spiking synchrony between neurons from the same cNE can be sufficiently explained by the degree of receptive field similarity between the neurons. Comparing STRF PWCs to spike train PWCs (*Figure 6A–6F*) revealed that the spike train PWCs (*Figure 6C*, histogram) were much narrower than the STRF PWCs (*Figure 6C*, solid line) for neuronal pairs within

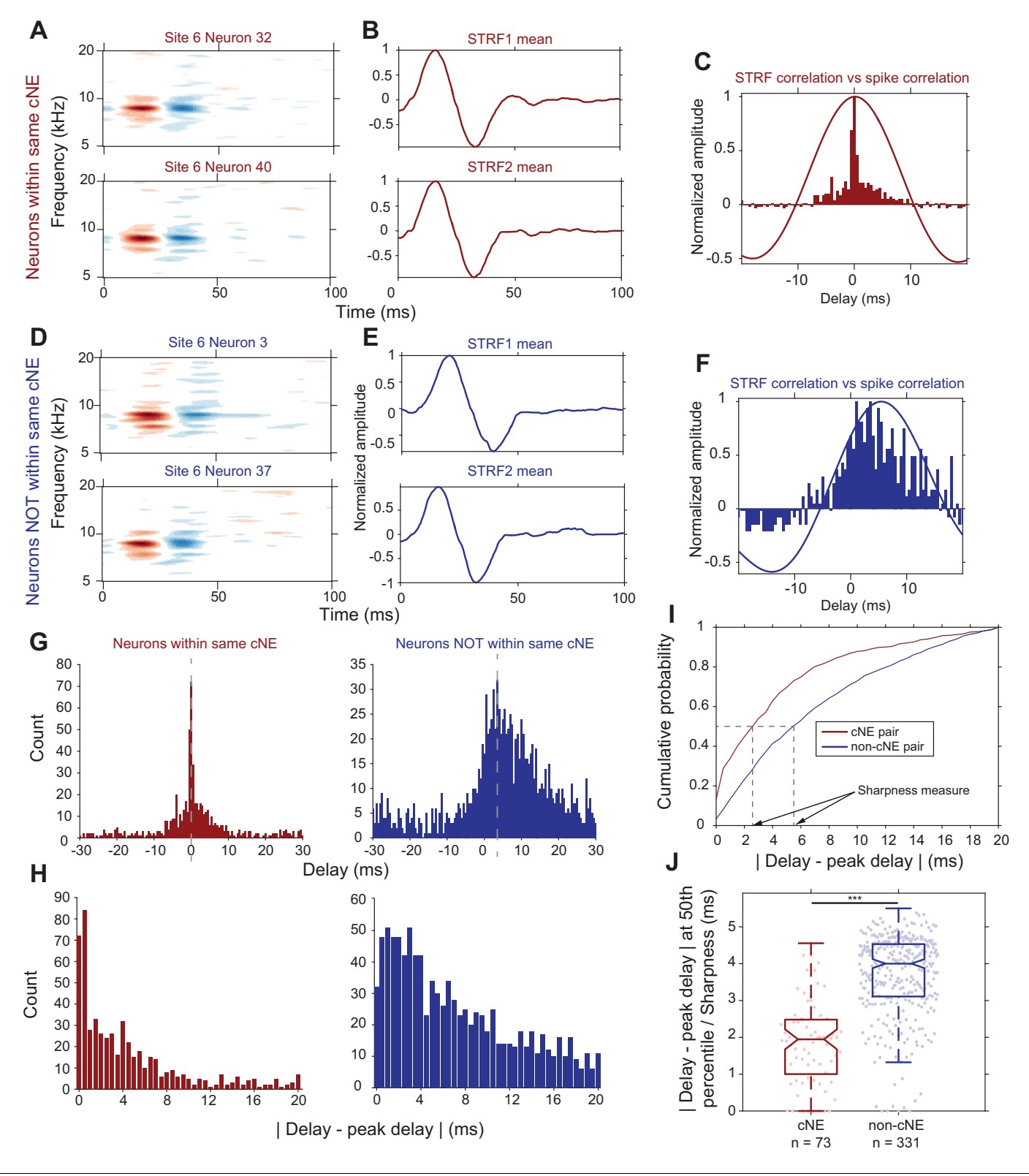

**Figure 6.** Synchrony between cNE members cannot be fully explained by receptive field overlap. (A–C) Sample data for two neurons within the same cNE. (D–F) Sample data for two neurons not within the same cNE (but from the same recording). (A, D) STRFs for the two pairs of neurons. (B, E) Mean temporal profiles of each STRF in (A, D), obtained by averaging across the frequency axis of STRFs at every point on the time axis. (C, F) Comparisons between STRF PWCs and spike train PWCs. Neurons within the same cNE exhibit sharper spike train PWCs than STRF PWCs. Neurons not within the

*Figure 6 continued on next page*

*Figure 6 continued*

same cNE exhibit spike train PWCs that are as wide as their STRF PWCs. (G–I) Method for calculating sharpness of a spike train PWC. (G) Non-normalized spike train PWCs from (C) and (F). Grey dashed lines mark the peak delay (peaks of PWCs). (H) PWCs folded around their peak delays (grey dashed lines in (G)). (I) CDFs of the distributions in (H). Black dashed lines represent the time from the peak delay at the median of the distribution, which are the sharpness values for the two example PWCs in (G). (J) Sharpness comparisons for one penetration. Spike train PWCs of neurons within the same cNE are sharper than those of neurons not within the same cNE. ***p < 0.001, Mann-Whitney U test.

DOI: https://doi.org/10.7554/eLife.35587.011

The following figure supplement is available for figure 6:

**Figure supplement 1.** Comparison of sharpness of PWC functions of pairs within the same cNE against that of pairs not within the same cNE for all datasets (other than the one shown in *Figure 6J*).

DOI: https://doi.org/10.7554/eLife.35587.012

the same cNE. In contrast, for non-cNE pairs, the spike train PWCs were often as wide as the STRF PWCs (*Figure 6F*).

To compare PWCs across the population, we assessed correlation sharpness by first estimating the peak delay (*Figure 6G*) and then folding the PWCs around that delay (*Figure 6H*). We estimated correlation sharpness as the width that accounted for half the spike count in the PWC histogram (*Figure 6I*). The PWCs were significantly narrower for cNE pairs than for non-cNE pairs (p < 0.001, Mann-Whitney U test; *Figure 6J*; see *Figure 6—figure supplement 1* for each of the other penetrations). The difference in sharpness between cNE pairs and non-cNE pairs was consistent with the high degree of correlation between PWC sharpness and spike train correlation (*Figure 2B*), the latter of which was used to define cNEs. The median PWC sharpness for member neurons was ~2 ms, reflecting the central peak of a highly synchronized portion of spikes (*Figure 6C and G*). However, the tail-portion of member PWCs do contain less-strictly synchronized spiking events, suggestive of a mix of highly and loosely synchronized events among cNE members.

Together, these observations indicate that spike train PWCs between members of the same cNE show strong evidence of strict synchrony that cannot be explained by receptive field overlap alone. PWCs between non-cNE pairs display loose synchrony that largely reflects coincidental activity expected from independent neurons with significant receptive field overlap.

## cNEs are not depth-biased and reflect local circuitry

Across 16 penetrations, we identified a total of 104 cNEs (*Figure 4*, *Figure 4—figure supplement 1*), obtained in response to either DMR or RN broadband stimuli. The average number of cNEs per penetration was 6.5 ± 2.0, but the higher the number of neurons isolated in a penetration, the higher the number of identified cNEs (*Figure 7A*). For each penetration, the number of cNEs was ~15% of the recorded number of neurons. The mean number of neurons in cNEs was 7.4 ± 2.5 neurons. This again depended on the number of isolated neurons (*Figure 7B*), with mean cNE size ~17% of the total number of recorded neurons in the column. However, the covariance between mean cNE size and the number of isolated neurons (*Figure 7B*; $R^2$ = 0.33) was not as strong as the covariance between the number of identified cNEs and the number of isolated neurons (*Figure 7A*; $R^2$ = 0.74). The IC weight of the vast majority of cNE member neurons was positive, that is, the neurons were co-activated with other members during instances of cNE events. However, 9 cNEs (8.3%) also contained at least one neuron with a negative weight, indicating that such neurons had to be inactive at the time of a cNE event. Of the 655 neurons that were isolated, a majority (~68%) belonged to a single cNE, ~9% did not belong to any cNE, and ~23% belonged to multiple cNEs (*Figure 7C*). A majority of pairs of neurons within the same cNE (~82%) tended to be recorded from channels separated by <200 μm (*Figure 7D*), indicating that cNEs are mostly spatially confined. These assessments were not biased by recording depth, since cNEs were found across the entire explored columnar range and showed no bias to any particular depth (*Figure 7E*). Most isolated cNEs (~70%) spanned a depth range of 200 μm or less of the cortical column, again indicating that cNE activity reflects local circuit properties, even though there were a few cNEs that spanned most of the measured cortical column (*Figure 7F*).

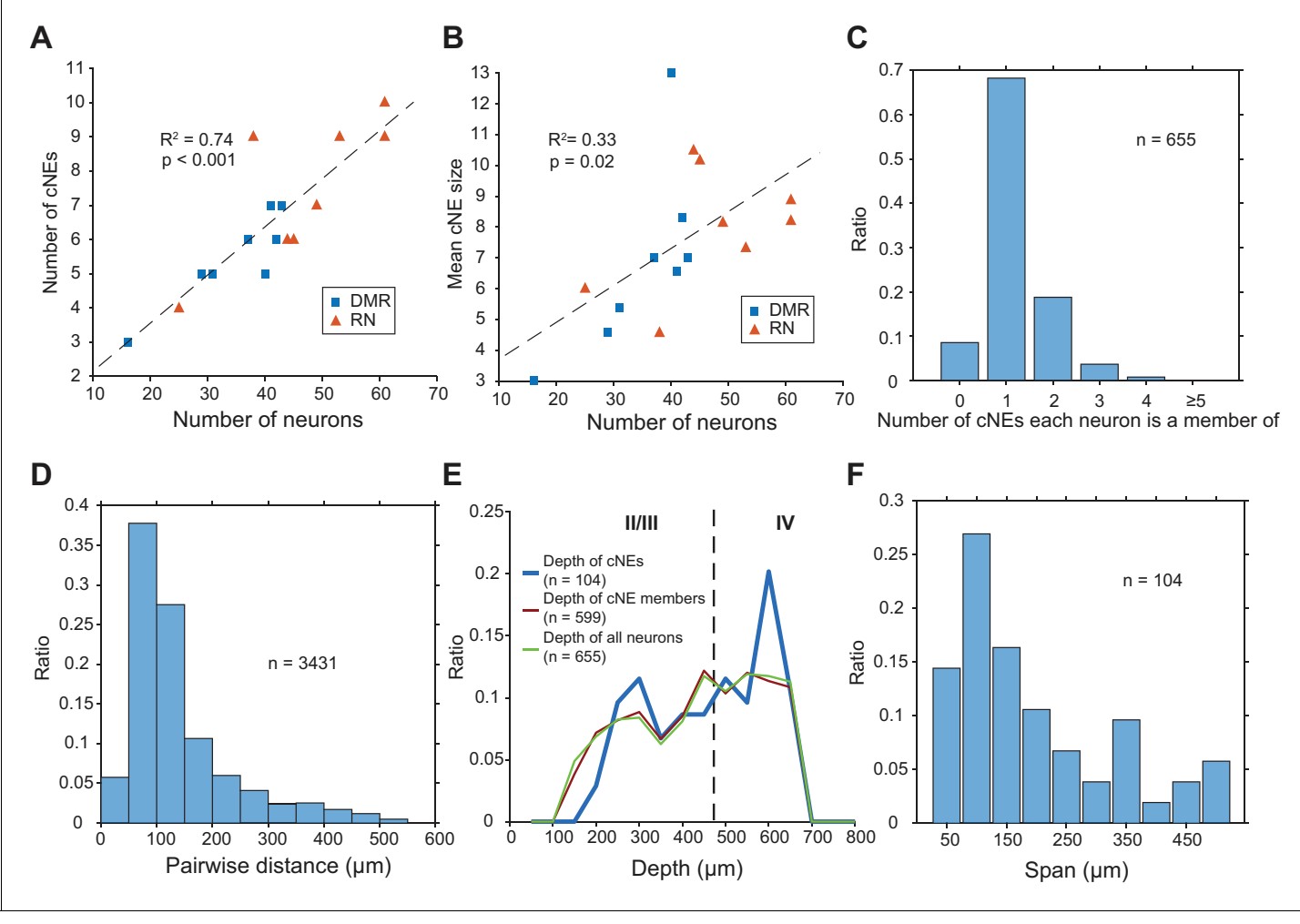

**Figure 7.** Properties of cNEs computed using 10-ms time bins. (A) Number of cNEs detected increases with the number of neurons recorded. (B) Mean cNE size increases with the number of neurons recorded. (C) Most neurons recorded (~68%) belong to only 1 cNE. A small subset of neurons (~9%) did not belong to any cNE. ~23% of neurons belonged to multiple cNEs. (D) Pairwise distances between neurons in the same cNE. Most pairs of neurons in the same cNE (~82%) were recorded within <200 µm of each other. (E) Depth of cNEs, of neurons within cNEs and of all isolated neurons. The depth of each cNE was calculated by taking the median depth of each cNE's member neurons. The depth of neurons within cNEs is the depth of all neurons found in at least one cNE, i.e. all recorded neurons except those found in the 0 bin in (C). The dashed line indicates the putative boundary between layers II/III and layer IV based on *Szymanski et al., 2009*. There was no depth bias for cNE neurons, that is the distributions of the depth of cNE members and that of all recorded neurons were similar. (F) cNE span, determined by the difference between the maximum and minimum depth of member neurons in each cNE. Most cNEs (~70%) had a span of 200 µm or less.

DOI: https://doi.org/10.7554/eLife.35587.013

## cNEs persist across epochs of spontaneous and evoked activity

Next, we wanted to determine if cNEs are dependent on stimulus-evoked activity or whether they are similar to cNEs obtained for spontaneous activity. Previous work in sensory cortices has shown that patterns of spontaneous activity are often similar to that of evoked activity (*Jermakowicz et al., 2009*; *Luczak et al., 2009*). To investigate if this is true of cNEs, we recorded contiguous epochs of spontaneous and evoked activity, separately processed the spontaneous and evoked data with the cNE detection algorithm, and compared the resulting spontaneous and evoked cNEs from each recording (*Figure 8A and B*). Using previously described methods (*Figure 4—figure supplement 2*), we defined a null distribution for each recording, and set the significance threshold at the 99th percentile (*Figure 8C*). Across all eight recordings comprising contiguous epochs of spontaneous and evoked activity we found significant matches for ~72% of spontaneous and evoked cNEs

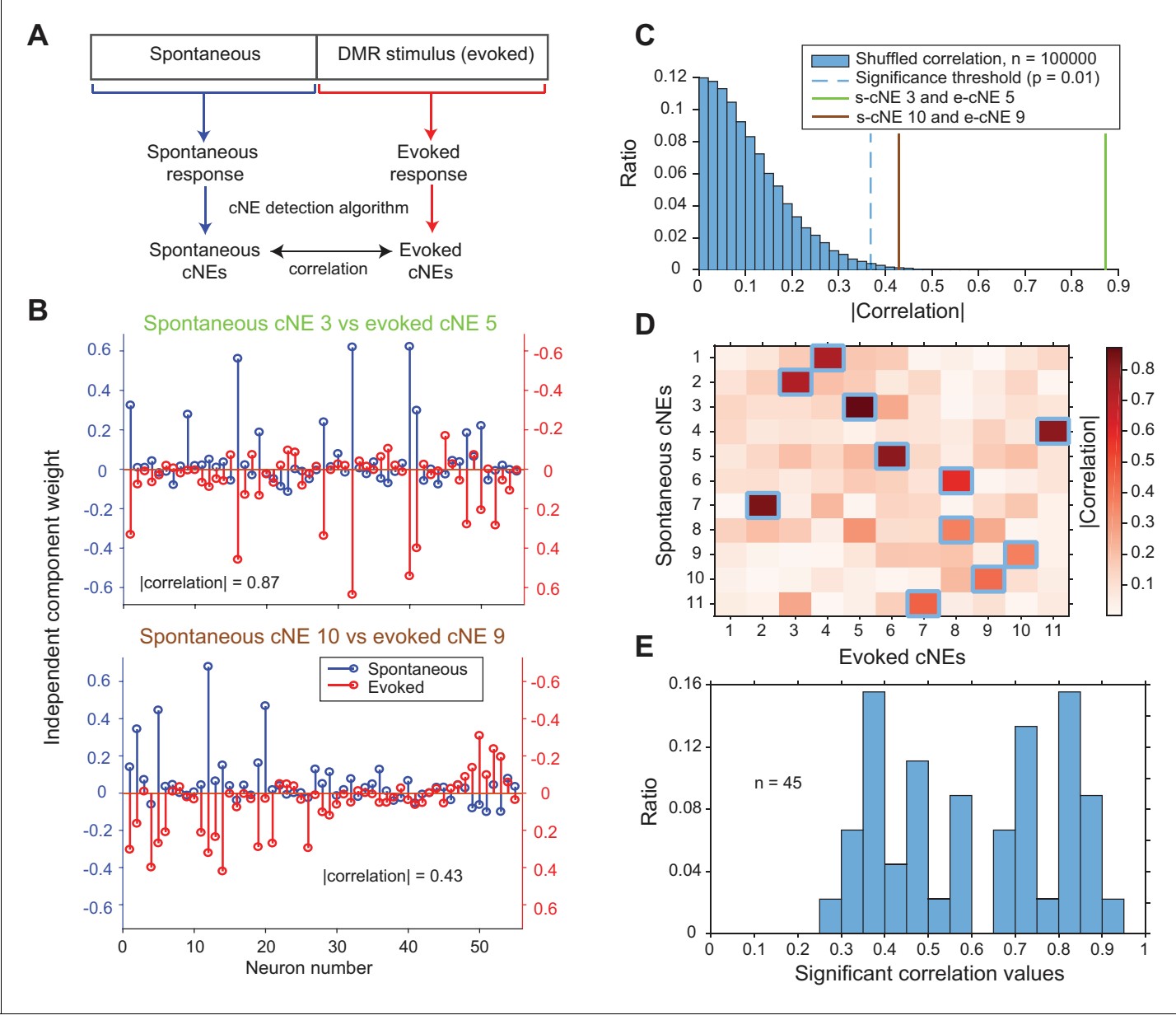

**Figure 8.** cNEs are highly preserved across spontaneous and evoked activity. (**A**) Illustration of the procedure for comparing spontaneous cNEs with evoked cNEs. (**B**) Examples of pairs of spontaneous-evoked cNEs from one penetration. (top) A pair of cNEs with an absolute correlation value of 0.87, implying that this particular cNE was highly preserved regardless of the presence of a stimulus. (bottom) A pair of cNEs with a lower absolute correlation value of 0.43. Almost all of the neurons with high weights were preserved and the low correlation value can be attributed to the noise in the neurons with low weights. (**C**) The two pairs of cNE examples in (**B**) were significantly matched. See text for how the null distributions (blue bars) were calculated. Setting a significant threshold of p = 0.01 for the null distribution (blue dashed line) revealed that both comparisons in (**B**) were significantly matched (green and brown solid lines). (**D**) Absolute correlation values between the weights of spontaneous and evoked cNEs in one penetration (that includes the two examples in (**B**)). Correlation values enclosed by blue squares indicate the correlation values that are higher than the significance threshold in (**C**). (**E**) Significant correlation values across all penetrations with contiguous spontaneous and DMR-evoked recordings. Approximately 72% of cNEs (both spontaneous and evoked) had significant matches.

DOI: https://doi.org/10.7554/eLife.35587.014

(*Figure 8D and E*). This implies that cNEs largely reflect functional connectivity and are not simply a reflection of stimulus-induced synchrony of neurons with overlapping receptive fields.

## cNEs show enhanced information processing

Stimulus-evoked synchronized spiking events can carry more information about the stimulus than the individual spikes of each member of the pair (*Atencio and Schreiner, 2013*; *Brenner et al., 2000*). To test whether cNE events also convey more stimulus information than the spikes of individual neurons that comprise each cNE, we assessed the mutual information (MI) for both types of events using the responses to fifty 5-s repeats of RN or DMR (*Brenner et al., 2000*, see Materials and methods). Intuitively, this method of determining MI is a proxy for the reliability of responses over stimulus trials. The lower the response reliability of repeated stimuli, the flatter the resulting PSTH, and the lower the reflected MI (*de Ruyter van Steveninck et al., 1997*; *Shih et al., 2011*; *Strong et al., 1998*). Spikes of each cNE member neuron were compared against the events of the cNE that each neuron belonged to (*Figure 9A and B*). cNE events were extracted by setting a threshold for the activity profile (*Figure 4E*, see Materials and methods). For each trial-based comparison between a cNE and one of its member neurons, the entity with the higher spike or event count was sub-sampled to maintain equal number of spikes or events and ensure unbiased MI comparisons. The cNE event rasters showed less trial-to-trial variability than individual neuron rasters (*Figure 9A and B*), and the MI conveyed by cNE events was, accordingly, ~10% higher than the MI of each member

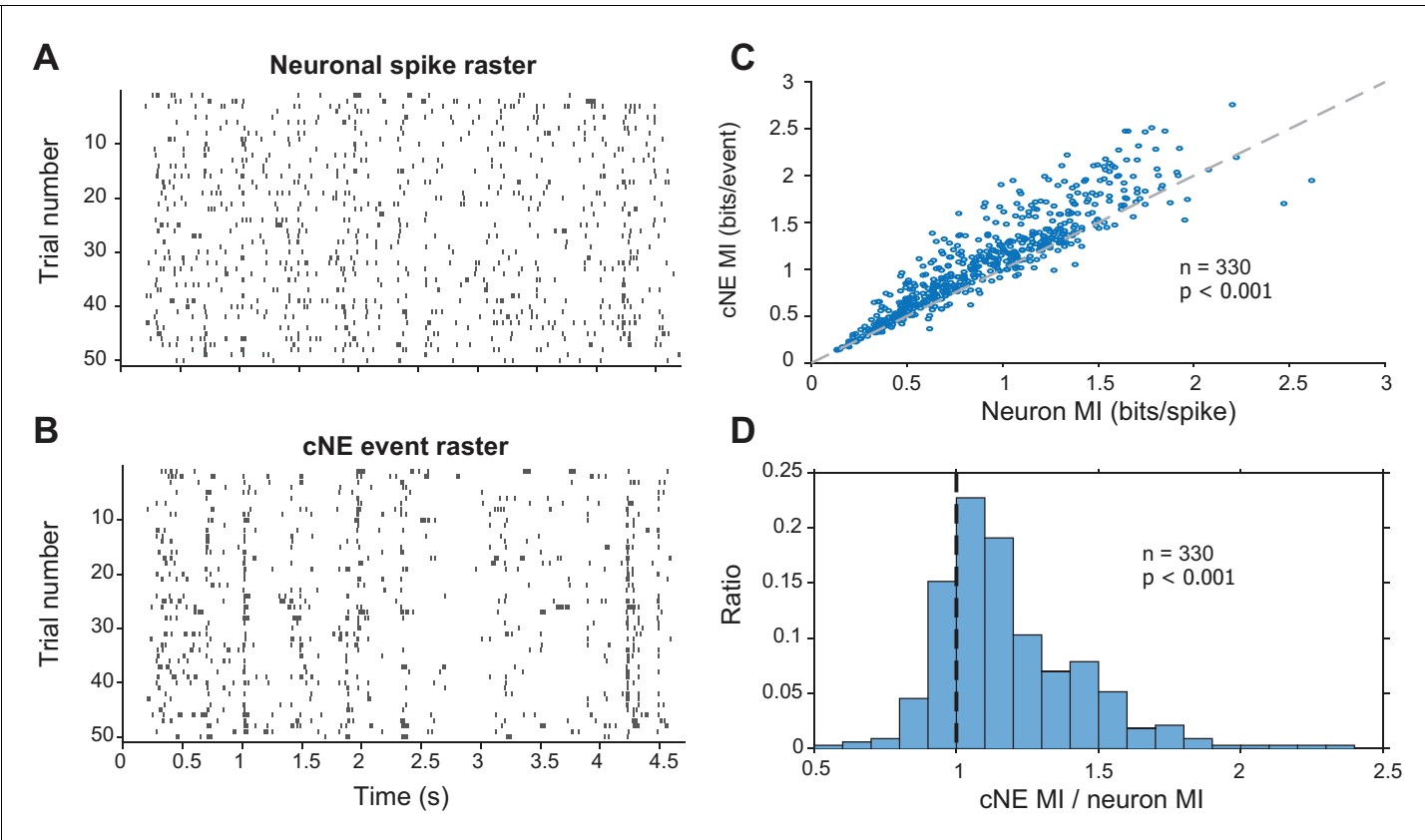

**Figure 9.** Mutual information (MI) carried by patterns of cNE events is higher than MI carried by patterns of single neuron spikes. (A) Raster for neuronal spikes over 50 presentations of a ~5-s long DMR or RN stimulus. (B) Raster for cNE events over 50 presentations of the stimulus. The neuron in (A) is a member of the cNE in (B). Random sampling was used to ensure equivalent spike or event counts for each comparison. (C) Population data for cNE MI against single neuron MI. cNE MI was significantly higher than that of the cNE's constituent neurons. (D) cNE MI/neuron MI. Wilcoxon signed-rank test was used in (C) and (D).

DOI: https://doi.org/10.7554/eLife.35587.015

The following figure supplements are available for figure 9:

**Figure supplement 1.** Mutual information (MI) carried by patterns of cNE members' spikes is higher than that of random groups of neurons.
DOI: https://doi.org/10.7554/eLife.35587.016

**Figure supplement 2.** Noise correlations between cNE pairs were higher than those between non-cNE pairs.
DOI: https://doi.org/10.7554/eLife.35587.017

neuron (p < 0.001, Wilcoxon signed-rank test; *Figure 9C*). Correspondingly, the ratio of the MIs (cNE/neuron) was significantly larger than 1 (median ± median absolute deviation = 1.11 ± 0.13; p < 0.001, Wilcoxon signed-rank test; *Figure 9D*).

Next, we established that the higher MI attributed to cNE events was not due to the fact that these cNE events integrated spikes across multiple neurons and, by virtue of a larger neuronal count, carried more MI. We therefore compared the MI for spikes of random groups of neurons (*Figure 9—figure supplement 1A*) to the MI for spikes of cNE members (*Figure 9—figure supplement 1B*). Each random group of neurons consisted of at least one neuron from the cNE (size N) that it was being compared against, along with N – 1 neurons that were pseudo-randomly selected from non-cNE members (i.e., neurons that were not part of the same cNE as the subject neuron; see Materials and methods). We found that the MI of spikes from cNE members was significantly higher than the MI of spikes from random groups of neurons (p < 0.001, Wilcoxon signed-rank test; *Figure 9—figure supplement 1D*). For each cNE-random group comparison, we also calculated a uniqueness index (UI), which quantifies the proportion of times the random group MI exceeded the cNE MI value, $\mathrm{UI} = \left(1 - \frac{x}{100}\right)$, where x is the percentile at which the cNE value was found in comparison to the random group MI distribution (*Figure 9—figure supplement 1C*; n = 501). The distribution of UI values was significantly skewed towards 0 and was significantly different from an expected uniform distribution that would be obtained if cNEs were composed of randomly selected neurons from population recordings (p < 0.001, Kolmogorov–Smirnov test). Together, these results show that cNEs respond more reliably to the stimulus than its member neurons, suggesting that they might have important functional roles in information processing and transmission in AI.

We were also interested to see if high noise correlations (for review, see *Cohen and Kohn, 2011*; *Kohn et al., 2016*) corresponded to neuronal pairs within detected cNEs. Over the 16 columnar recordings, pairs of neurons that were members of the same cNE had significantly higher noise correlations than pairs of neurons not within the same cNE (*Figure 9—figure supplement 2*). This is consistent with common synaptic input being one potential source of noise correlations (*Kanitscheider et al., 2015*; *Kohn and Smith, 2005*; *Shadlen and Newsome, 1998*). Since neurons in the same cNE have reliable synchronous activity, it is likely that they receive more similar common synaptic inputs than neurons not within the same cNE.

To determine the kind of information being enhanced by cNEs, we compared the STRFs (calculated via spike-triggered or event-triggered averages) of cNEs and their member neurons. Spike or event trains were subsampled so that they had equal numbers of spikes or events. cNE STRFs had stronger excitatory and inhibitory subfields than their member neurons (*Figure 10A*). Compared to neuronal STRFs, cNE STRFs had higher peak-trough differences (PTD; p < 0.001, Wilcoxon signed-rank test; *Figure 10B*) and MI (p < 0.001, Wilcoxon signed-rank test; *Figure 10C*). Thus, cNE STRFs conveyed more information about the stimulus than neuronal STRFs by increasing the signal-to-noise ratio represented in their STRFs.

To show that the increase in information conveyed by cNE STRFs was not trivially due to the fact that cNEs integrate over multiple neurons with similar STRFs and, thus, must convey more information, we also compared multi-unit STRFs of cNE member neurons against multi-unit STRFs of random group of neurons (*Figure 10—figure supplement 1A*). Across the entire population, the STRF PTD for cNE members was significantly higher than that of the STRF PTD for random groups of neurons (p < 0.001, Wilcoxon signed-rank test; *Figure 10—figure supplement 1C*). The STRF MI of cNE member neurons was also significantly higher than that of random groups of neurons (p < 0.001, Wilcoxon signed-rank test; *Figure 10—figure supplement 1E*). For both STRF PTD and MI, we found that the distribution of UI values was significantly skewed towards 0, and was significantly different from a uniform distribution, which would be obtained if cNEs were composed of randomly selected neurons from population recordings (p < 0.001, Kolmogorov–Smirnov test).

Taken together, the MI results (*Figures 9* and *10* and their figure supplements) suggest that cNEs enhance information processing by reducing the number of stimulus-independent events or increasing the ratio of stimulus-dependent events. This is reflected in the increase in response reliability and magnitude of the excitatory and inhibitory subfields in the cNE STRF compared against its member neurons.

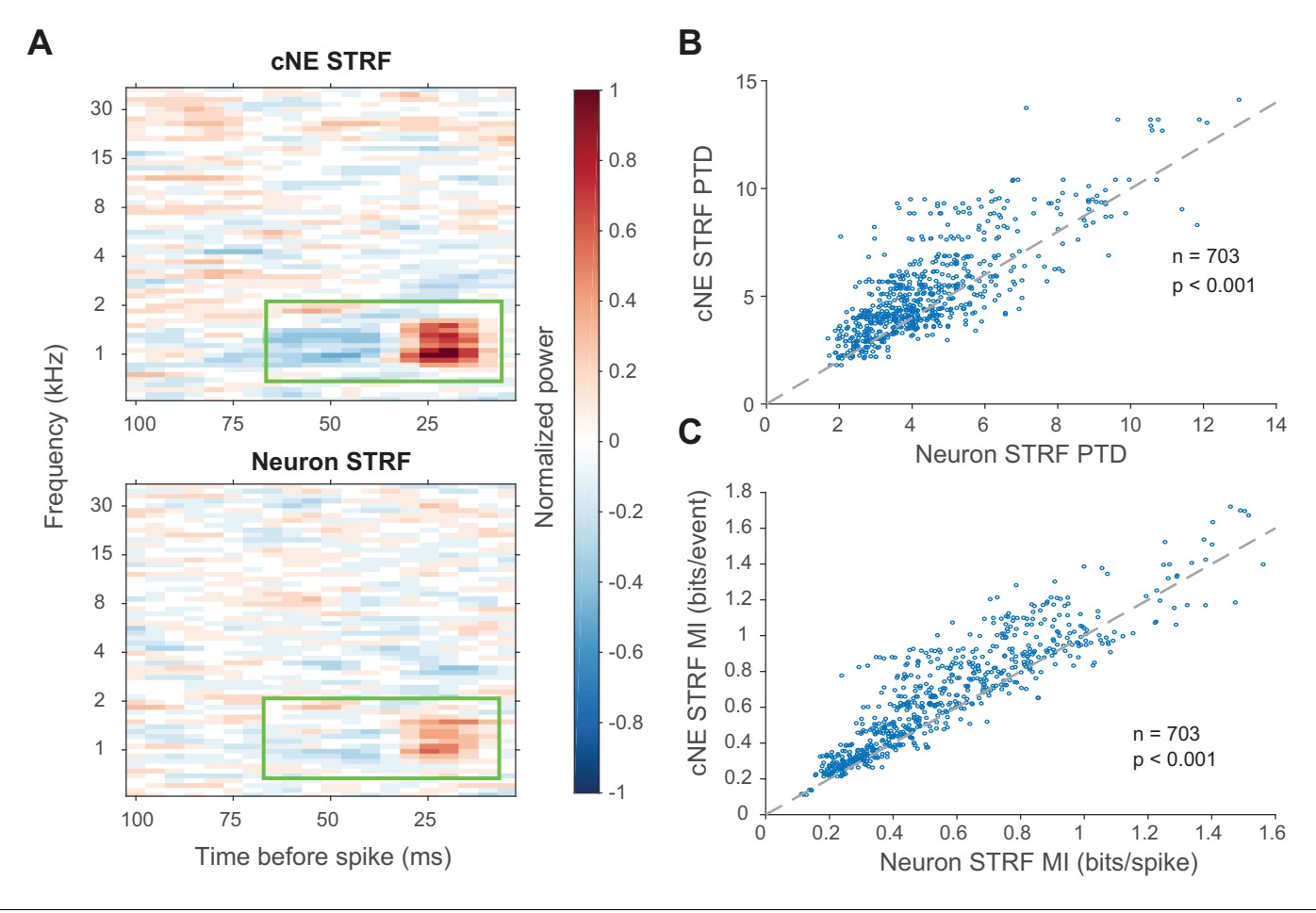

**Figure 10.** cNE STRFs have enhanced features compared to STRFs of member neurons. (A) Example comparison between a cNE's STRF (top) and a member neuron's STRF (bottom). Random sampling was done to estimate STRFs with equivalent spike/event counts. The color scales for the sample STRFs have been normalized. The cNE STRF has stronger excitatory and inhibitory subfields than the neuronal STRF (highlighted by green boxes). (B) Over the entire population, cNE STRFs had a higher peak-trough difference (PTD) than neuronal STRFs. PTD was the difference between the highest and lowest STRF values, divided by the number of spikes/events. (C) Similarly, cNEs had higher mutual information (MI) between their STRFs and single events than neurons had between their STRFs and single spikes. Wilcoxon signed-rank test was used in (B) and (C).

DOI: https://doi.org/10.7554/eLife.35587.018

The following figure supplement is available for figure 10:

**Figure supplement 1.** cNE members have multi-unit STRFs with enhanced features compared to the multi-unit STRFs of random groups of neurons.

DOI: https://doi.org/10.7554/eLife.35587.019

## cNE events are not fully accounted for by stimulus synchronization or receptive field overlap

Finally, we investigated whether the coordinated responses observed in detected cNEs can be accounted for by stimulus-driven response synchronization resulting from either receptive field overlap of cNE member neurons or by second-order (pairwise) correlations. First, we repeated a 10-min ripple noise stimulus 15 times and examined units that responded to each stimulus repetition with firing rates that varied by less than 30% across adjacent stimulus presentations. We isolated 24 units that satisfied these criteria (*Figure 11—figure supplement 1*). For the simultaneously recorded ('real') spike trains, we randomly selected 15 of the 24 neurons from one repetition at a time and calculated cNEs (for a total of 100 unique combinations). For the non-simultaneous ('surrogate') spike train matrix, we selected the same 100 unique combinations of 15 neurons, but the spike train of each neuron was extracted from a different stimulus repetition (*Figure 11—figure supplement 1*). If the cNEs had resulted purely from stimulus-driven synchronization, the simultaneous/'real' spike train

matrices should have cNE statistics that were similar to that of the non-simultaneous/'surrogate' spike train matrices. We found, however, that the 'surrogate' spike train matrices had fewer cNEs (*Figure 11A*) and that the detected cNEs had fewer member neurons (*Figure 11B*). This suggests that even though some stimulus-driven synchronization, including synchronization by features of the stimulus not represented in STRFs, is present in the non-simultaneous spike trains, it is not the dominant contributor to synchrony observed in the simultaneous recordings.

We next modeled spike trains to confirm that receptive field similarities among cNE members and second-order correlations did not account for the high degree of observed cNE synchrony. In a first model ('shuffled'), we circularly shuffled each of the simultaneously recorded spike trains, breaking the temporal relationships between neurons as well as the match between stimulus and receptive field at each spike's occurrence while preserving inter-spike intervals (ISIs) for each neuron (see Materials and methods).

In a second model ('preserved receptive field' or 'PR'), we again shuffled the spike times of neurons, but in a way that preserved the match between stimulus and receptive fields at the new spike times (*Figure 11—figure supplement 2A*). To achieve that, we calculated the similarity between stimulus segments and neuronal STRFs (projection value) and shuffled spikes between time bins with similar projection values (*Figure 11—figure supplement 2A*, see Materials and methods). This procedure created surrogate spike trains that had shuffled spike times but similar receptive fields to the original spike trains.

In the third model, we used a parametric analysis to investigate if second-order correlations were sufficient to describe the degree of synchrony we see in cNE members. We used a 'dichotomized Gaussian' ('DG') model (*Macke et al., 2011*; *2009*, see Materials and methods), which creates spike trains that match both the firing rate of each neuron and the PWC between each pair of neurons. Thus, this model accounts for the overall activity of neurons up to second-order correlations but does not contain information about higher-order correlations.

To validate that the models behaved as expected, we calculated the STRFs for each neuron in each dataset via spike-triggered averaging. The 'PR' model preserved STRFs relative to STRFs derived from the real spike trains (*Figure 11—figure supplement 3A*), indicating that the long-term receptive field information was accurately preserved. As expected, the 'shuffled' and 'DG' models did not provide well-defined STRFs (*Figure 11—figure supplement 3A*). When STRF similarity between the three different models and the real spike train was calculated, the 'PR' model had a high median correlation value (~0.74) while the 'shuffled' and 'DG' models had values close to zero (*Figure 11—figure supplement 3B*). We also calculated PWCs for pairs of neurons in the real spike train and the models and found that the 'shuffled' and 'PR' models captured very little of the correlation between pairs of neurons in the real spike train, while the 'DG' model only captured the correlation between pairs of neurons in the real spike train at zero delay (*Figure 11—figure supplement 3C*). This is expected because the 'shuffled' model breaks up all inter-neuronal correlations; the 'PR' model only preserves receptive fields, and PWCs between the temporal profiles of STRFs are broad (*Figure 6*); and the 'DG' model is based on correlations between pairs of neurons. Since the correlation value is a single value estimated across the entire spike train, it does not take into account intra-spike train synchrony and does not model delay-dependent temporal correlations. However, since PWCs between pairs of neurons in the same cNE are sharp, with mean sharpness values of 3.39 ± 1.21 ms (*Figure 6G–6J*, *Figure 6—figure supplement 1*), and mean absolute peak delays of 0.76 ± 1.74 ms, modeling spike trains using the 'DG' model at 5-ms temporal resolution effectively encompasses the temporal correlations between neurons from the same cNE (*Figure 11—figure supplement 3C*).

Surrogate spike trains from the 'shuffled' and 'PR' models had significantly fewer cNEs (*Figure 11C*) than the real data. On the other hand, the 'DG' model overestimates the number of detected cNEs (*Figure 11C*). Despite that, all three models, including the 'DG' model, resulted in cNEs that were smaller in size (i.e., fewer member neurons, *Figure 11D*) than the real data (n = 16 penetrations; mean values for each penetration were calculated over 200 iterations for each model). These results imply that the observed synchrony represents unique network events that cannot be explained simply by second-order correlations or by synchrony reflective of the underlying similarity in the receptive fields of neurons.

Finally, to estimate the degree of coincident firing between cNE member neurons, we calculated coincidence ratios (CRs) between groups of neurons. The CR was defined as the ratio of the number

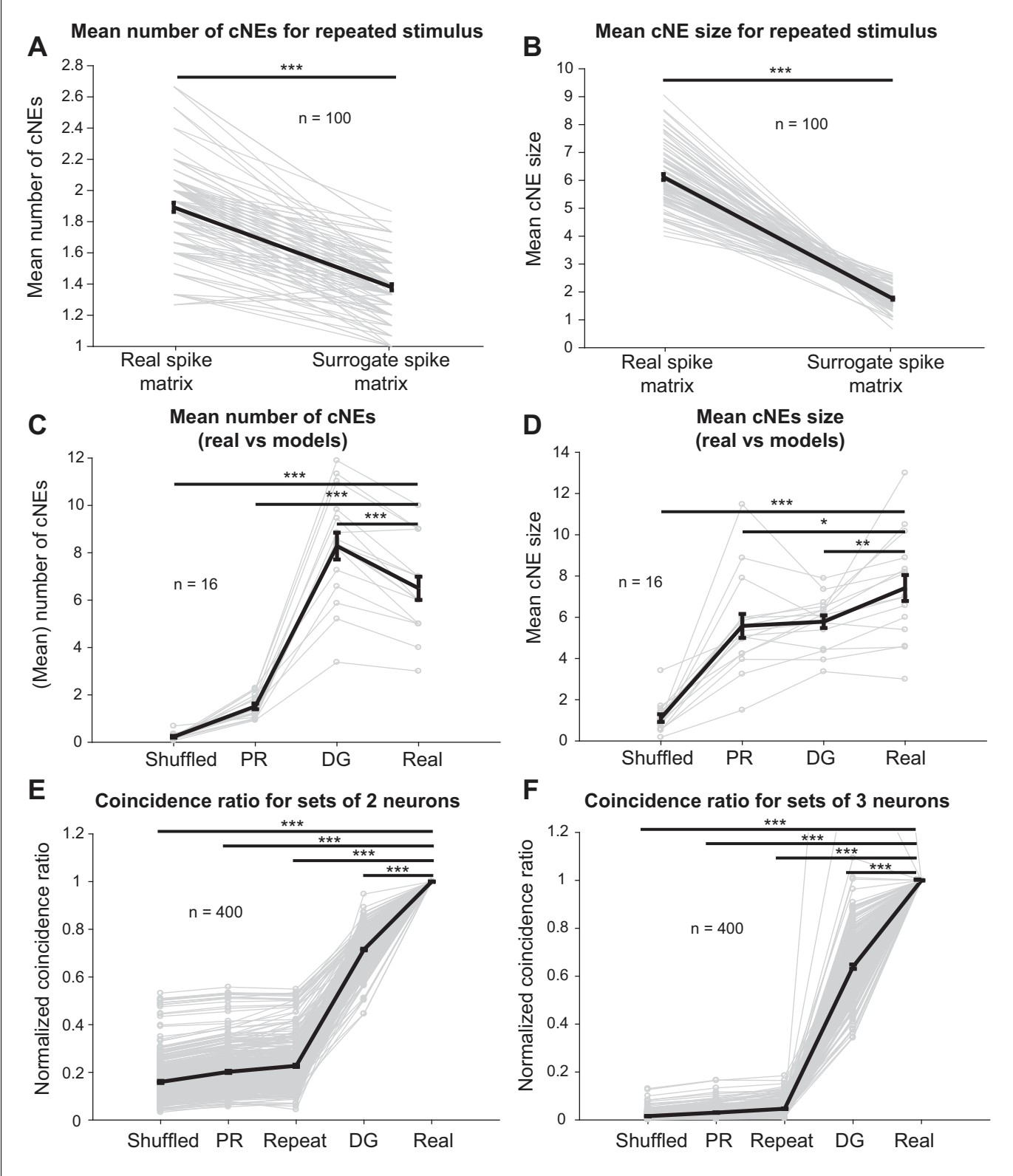

**Figure 11.** Receptive field similarity is insufficient to explain the coincident spiking between neurons in the same cNE. (A) The number of cNEs identified in 'real' spike train matrices is significantly higher than that of 'surrogate' spike train matrices. (B) Mean size of cNEs identified in 'real' spike train matrices is significantly larger than that of 'surrogate' spike train matrices. See *Figure 11—figure supplement 1* for details on how the spike train matrices were constructed. Each paired dataset represents one unique combination of 15 neurons. (C) The number of cNEs identified in the 'real' data

*Figure 11 continued on next page*

*Figure 11 continued*

is significantly higher than that of the 'shuffled' and 'PR' models, but is significantly lower than that of the 'DG' model. (D) The mean size of cNEs identified in the 'real' data is significantly larger than that of the 'shuffled', 'PR' and 'DG' models. See *Figure 11—figure supplement 2A* for how the 'PR' model was computed. Each dataset represents the mean number of cNEs and the mean cNE size for one recording over 200 iterations. (E, F) The coincidence between neurons from different repeats of the same stimulus ('repeat'), or from the 'shuffled', 'PR' and 'DG' models is significantly less than the coincidence seen between neurons from the same cNE in the 'real' data. See *Figure 11—figure supplement 4* for how the coincidence ratio (CR) was calculated. Each dataset represents a unique combination of 2 or three neurons and was normalized by the CR between neurons from the 'real' spike matrix. Sets of more than three neurons had coincidence ratios that were close to zero for most models and are not shown. *p < 0.05, **p < 0.01 ***p < 0.001, paired t-test with Bonferroni correction.

DOI: https://doi.org/10.7554/eLife.35587.020

The following figure supplements are available for figure 11:

**Figure supplement 1.** Illustration of the process of constructing spike trains from different presentations of the same stimuli to obtain 'real' and 'surrogate' spike matrices.

DOI: https://doi.org/10.7554/eLife.35587.021

**Figure supplement 2.** Method for the 'preserved receptive field' ('PR') model.

DOI: https://doi.org/10.7554/eLife.35587.022

**Figure supplement 3.** Verification of the 'shuffled', 'PR', and 'DG' models.

DOI: https://doi.org/10.7554/eLife.35587.023

**Figure supplement 4.** Calculation of coincidence ratio between groups of neurons in the same cNE.

DOI: https://doi.org/10.7554/eLife.35587.024

of time bins in which all neurons in the pre-defined group spike together, to the number of time bins in which the neuron with the lowest firing rate is active. When the whole group is active each time the neuron with the lowest firing rate spikes, the CR value will be one (*Figure 11—figure supplement 4A*). This measure allows us to compare the probability that several neurons in a cNE will fire together and can be thought of as a generalization of the dot product between two spike trains to multiple spike trains. Using this measure, we directly compared our responses across different presentations of the same stimulus and the simulated spike trains (1000 iterations each) to the real data (*Figure 11E and F*), by estimating CRs for 400 unique groups of 2, 3, 4, and 5 neurons within the same cNE (*Figure 11—figure supplement 4B*). Since neuronal identity was preserved in all simulations and the repeated stimulus data, we normalized each group of neurons by the CR of the real data, to determine how well the synchrony from simulated data approximated the synchrony from the real data. The normalized CR for the repeated stimulus data ('repeat'), 'shuffled', 'PR' and 'DG' models were significantly less than one for all groups (*Figure 11E and F*). For groups containing more than three members, the CR values for most control models were always near or at 0 (not shown). These results suggest that member neurons of cNEs have more coincident activity than expected, further supporting the hypothesis that coordinated cNE events are driven by higher-than-second-order correlations and cannot be explained simply via stimulus-induced synchronization or shared receptive field properties, including those not reflected in STRFs.

## Discussion

A key goal of this study was to test the hypothesis that auditory processing of an ongoing stimulus is better interpreted as sequences of coordinated activity among small subsets of neurons in the cortical column, rather than independent spikes distributed across the same subsets of neurons. We were able to identify, for each columnar recording in rat AI, several groups of neurons with coordinated activity (cNEs) in response to broadband stimuli with dynamic spectro-temporal envelopes. Each identified cNE comprised a subset of ~15–20% of the simultaneously recorded neurons. The spatial distribution of neurons within most cNEs was contained within <250 μm of columnar depth, approximately corresponding to the width of granular and supragranular layers. However, several cNEs encompassed neurons distributed across multiple layers.

We isolated and identified groups of neurons in AI columns that showed coordinated spiking activity by using dimensionality-reduction techniques (*Lopes-dos-Santos et al., 2013*). The group spiking events associated with each cNE were more reliable and conveyed more information about the stimulus than spikes of individual neurons or random groups of neurons (matched to cNE size).

We verified that identified cNEs were not chance constructs as a result of stimulus-driven synchronization alone by showing that the frequency of occurrence of cNE events was higher than predicted from basic receptive field similarities within the column. Since a high percentage of cNEs had matching neuronal composition during evoked and spontaneous activations, these findings support the concept that cNEs are stable local units of information processing in the cortical column.

## Challenges in detecting cNEs in AI

Detecting or identifying cNEs in sensory cortex is difficult due to the multidimensional nature of population recordings. The first difficulty is implementing a method that may estimate the number and members of ensembles from high-dimensional data. Another difficulty is that any method will rely on responses occurring over chosen time windows and may be confounded with the temporal information processing performed by the region of interest. Non-auditory responses are more easily examined for ensemble activity because stimuli that drive the neurons in these regions are not functions of time. While the visual scene may vary over time, it is not dominated by rapid temporal information. Since visual images are predominantly static, visual cortical studies have used two-photon calcium imaging, which measures a response signal over an integration window of 125–250 ms (*Figure 5—source data 1*; *Miller et al., 2014*; *Carrillo-Reid et al., 2015*). Such imaging approaches and time windows cannot be utilized in the auditory cortex, since auditory cortical neurons are responsive to envelope modulations occurring over tens of milliseconds. Only static stimuli, such as pure tones or spectral ripple stimuli, could be utilized for auditory imaging studies. Thus, it is uniquely challenging to identify ensembles in auditory cortical regions due to the confound between analysis windows and temporal stimulus preferences.

Once estimated, interpreting auditory cortical ensembles poses additional challenges. Visual cortical receptive fields are often estimated over long time delays, while auditory cortical receptive fields based on dynamic stimuli necessarily contain more precise timing information. Thus, ensemble synchrony might be due to either stimulus-driven receptive field similarity, or it may be due to specific network configurations that favor joint activity, such as high anatomical or functional connectivity. To determine if ensembles represent more than the consequences of receptive field similarity, we utilized a variety of controls and showed that while ensemble activity can be related to stimulus preferences, it cannot be wholly explained through similarities in basic receptive field processing (*Figures 6* and *11*). Furthermore, the observation that spontaneous activity engages local networks that are very similar to those engaged by externally driven activity (*Figure 8*) supports the idea of the utilization of basic network building blocks (*Hebb, 1949*) in the transition from representation to interpretation to behavior. cNEs appear to be stable, functionally connected subsets of neurons that are embedded in broader columnar and horizontal networks whose activity reliably signifies specific auditory information.

## Detection and identification of cNEs

Even though technologies for recording large populations of neurons have been growing exponentially (*Blanche et al., 2005*; *Du et al., 2011*; *Rios et al., 2016*; *Stevenson and Kording, 2011*), the ability to make meaningful observations about network activity or ensembles of neurons has been hampered by the lack of analytic tools to detect and identify such constructs. New methods have more recently been proposed to identify functionally meaningful groups or ensembles of neurons (*Billeh et al., 2014*; *Gourévitch and Eggermont, 2010*; *Lopes-dos-Santos et al., 2013*; *Miller et al., 2014*; *Peyrache et al., 2010*; *Pipa et al., 2008*), either as neurons that fire synchronously within a predefined time window (*Harris et al., 2003*), or as neurons with fixed sequential firing patterns over time, akin to synfire chains (*Ikegaya et al., 2004*). Columnar AI responses to stimuli (*Atencio and Schreiner, 2010a*; *Kaur et al., 2005*) or thalamocortical excitation (*Krause et al., 2014*) have been previously shown to occur within a narrow time window (<10 ms), and we have defined groups of cooperating neurons as neurons that spike synchronously within a 10-ms time window. Previous studies have demonstrated that the analysis of (near) zero-lag synchrony can provide rich insights into network function (*Atencio and Schreiner, 2013*; *Harris, 2005*; *Nicolelis et al., 1995*), and is more easily identified by statistical methods than when considering sequential firing patterns over time (*Peyrache et al., 2009*). Finally, if we consider that cNEs are dynamically assembled, and that connectivity between cNE members can change over time

(*Buzsáki, 2010*; *Reimann et al., 2017*), it is essential to follow them at rapid time scales, which is intractable if large time lags are considered.

We made use of dimensionality reduction techniques, namely PCA and ICA (*Lopes-dos-Santos et al., 2013*), to detect cNEs. This approach extracts the number of significant cNEs, assigns weights according to each neuron's contribution to each cNE, and gives an intuitive measure of cNE activity. The identification of significant cNEs was also calculated using established results from random matrix theory (*Marčenko and Pastur, 1967*), and therefore reduces the reliance on surrogate randomization methods. The method is mostly linear; ICA uses nonlinear equations to quantify negentropy (a measure of distance between a distribution and the Gaussian distribution), but this is highly optimized by the fast ICA algorithm (*Hyvärinen and Oja, 1997*), making it significantly faster than most nonlinear algorithms. Since we were focusing on the cortical column, we limited our examination to synchronous spiking within 10-ms time windows. The short time window that we used deemphasizes the need to assess neuronal ensembles as a function of the lag between time bins (*Russo and Durstewitz, 2017*). Furthermore, since our neuronal firing rates are approximately stationary, we did not correct for nonstationarity (*Russo and Durstewitz, 2017*).

## cNEs are distinct from cell assemblies or simple populations of neurons

Because we defined cNEs via synchronous firing within temporal bins, we termed these constructs 'coordinated neuronal ensembles' instead of the more commonly used term 'cell assemblies'. Cell assemblies, as first defined in Donald Hebb's seminal work (*Hebb, 1949*), were hypothesized to be fixed units of neurons that are highly and strongly interconnected, and that the activation of a few of its members would be sufficient to reliably cause the activation of the whole assembly. Hebb's cell assemblies also relate to large scale behavioral phenomenon (*Hebb, 1949*) and have been proposed to require a read-out mechanism (*Buzsáki, 2010*). Since our groups of neurons have not been linked to global aspects of behavior, and do not include the activity of potential target neurons outside the column, we chose to use the term 'coordinated neuronal ensembles' to distinguish it from the more functionally related term 'cell assemblies'.

Another distinction is that the identity of neurons contributing to each cNE is important and their roles should be distinguished from neurons accumulated for the purpose of deriving simple population codes (*Bizley et al., 2010*; *Chakraborty et al., 2007*; *Furukawa et al., 2000*; *Rodgers and DeWeese, 2014*). We showed that neurons classified as belonging to the same cNE were significantly more synchronous with one another than those from simultaneously recorded non-members (*Figures 6* and *11*). We also showed that identified cNEs tended to convey more information about the stimulus than simultaneously recorded, arbitrary groups of neurons of the same size (*Figure 9—figure supplement 1* and *Figure 10—figure supplement 1*). Altogether, we demonstrated that cNEs are a subset of simultaneously recorded neurons that have sharply synchronized activity with enhanced information processing abilities.

## cNE activity does not require auditory stimuli

We demonstrated that the majority of cNEs (~72%) for spontaneous and evoked activity were significantly matched in member neuron composition (*Figure 8*). This implies that cNEs are stable functional entities that reflect stimulus selectivity but do not require external stimuli to be activated. This also supports previous findings in sensory cortices that patterns of spontaneous activity can be similar to those of evoked activity (*Jermakowicz et al., 2009*; *Luczak et al., 2009*), albeit at much finer timescales. These underlying functional network units have also been hypothesized to reflect constraints that cortical organization has imposed on the local propagation of neuronal activity (for in-depth reviews see *Luczak et al., 2015*; *Luczak and Maclean, 2012*).

The identity and stability of cNEs across spontaneous and evoked activity has only been established over relatively short periods of time (usually less than an hour under anesthesia). Previous studies have shown that exposure to specific stimuli tended to bias spontaneous activity towards the firing patterns seen in activity evoked by those specific stimuli (*Bermudez Contreras et al., 2013*; *Eagleman and Dragoi, 2012*; *Han et al., 2008*). This suggests that cNEs might similarly be mutable by dominant aspects of the auditory environment. By extension, we speculate that cNE composition and functional identity can be influenced by learning or any plasticity-inducing experiences. These potential effects need to be addressed in future work.

## cNEs cannot be accounted for by stimulus-driven synchronizations

A fundamental question regarding the identified cNEs is whether they were constructs that arose from the trivial fact that their member neurons all had similar receptive fields and, therefore, were simultaneously driven by the same stimulus. AI neurons in a column have been previously shown to have fairly closely related STRFs (*Atencio and Schreiner, 2010a*; *2010b*; *Guo et al., 2012*; *Wallace and Palmer, 2008*), making stimulus-driven synchronization a likely contributor to the observed synchrony of cNEs.

We demonstrated, however, that cNE membership was not solely accounted for by receptive field overlap using several independent methods. Spike train PWCs between pairs of neurons in the same cNE are significantly narrower than that of STRF PWCs (*Figure 6*). Assuming that pairs of neurons are simply synchronized by the stimulus, it follows that the spike PWCs would have similar widths to that of their receptive field PWCs. This is often true for pairs of neurons not within the same cNE (*Figure 6D–6F*), putative examples of neurons that are only synchronized due to their receptive field overlap.

Moreover, we showed that cNEs detected during evoked activity were often also detected during spontaneous activity (*Figure 8*). Consequently, the evoked cNEs that had significant matches to spontaneous cNEs did not require input synchronization by a stimulus.

Another argument against stimulus-driven synchrony as the main reason for the observed synchrony is the lower degree of synchrony between independently firing neurons than that of the coordinated firing of cNE neurons (*Figure 11—figure supplement 1*). The reduced synchrony, expressed in terms of lower cNE numbers and smaller cNE sizes for non-simultaneous spike matrices ('surrogate' spike matrices; *Figure 11A and B*), indicates that basic receptive field similarity and overlap are insufficient to account for the observed coordinated firing patterns.

Finally, we modelled stimulus-driven synchrony with the 'preserved receptive field' ('PR') model (*Figure 11—figure supplement 2A*). The idea behind this model was to shuffle the spike times of each neuron only among time bins associated with the same degree of similarity between the stimulus and receptive field (projection values). Given the probabilistic nature of neuronal spiking even for the highest projection values, the 'PR' model had, unsurprisingly, a lower number of cNEs per spike matrix and fewer neurons per cNE (*Figure 11C and D*). The CR for the 'PR' model was also significantly less than that of the real data for all sizes of sets of neurons (*Figure 11E and F*). These observations further support the notion that multi-neuronal coordinated activity requires conditions beyond receptive field overlap. The linear STRFs used here are, of course, not a complete representation of the stimulus preference of a neuron. Additional stimulus features may activate neurons, often in nonlinear fashion, as has been demonstrated using multi-filter approaches (*Atencio et al., 2008*; *Harper et al., 2016*). Even though the main (linear) filters appear to carry the most information, we cannot completely exclude the possibility that shared but 'hidden' receptive field properties, not reflected in either basic or advanced multi-filter STRFs, contribute to the formation and activation of cNEs. However, these contributions cannot be very substantial, as indicated by our model-free analysis of cNE synchrony (*Figure 11A, B* and *Figure 11—figure supplement 1*). Further determinants of cNE events are likely governed by more globally defined network states, including top-down influences (*Harris and Thiele, 2011*; *McGinley et al., 2015*; *Okun et al., 2010*).

## cNEs reflect higher-order correlations

To test if second-order correlations were sufficient to account for the synchrony observed between cNE neurons, we used the 'DG' model (*Figure 11C–11F*). The 'DG' model is fully specified by neuronal firing rates and the covariance of actual spike trains. It produces a set of spike trains that reproduces these statistics, but does not contain information about higher-order correlations (*Macke et al., 2011*; *2009*, see Materials and methods). The 'DG' model was able to capture correlations between all possible pairs of neurons but was unable to directly replicate the correlations between three or more neurons (*Figure 11—figure supplement 3C*). The lack of higher-order correlations resulted in smaller cNE sizes (*Figure 11D*) and lower CRs for three or more neurons that were members of the same cNE (*Figure 11F*). The lower CR for pairs of neurons from the same cNE (*Figure 11E*) shows that the 'DG' model incompletely captures the columnar synchrony of AI neurons (*Figure 11—figure supplement 3C*).

That second-order (pairwise) correlations were unable to recapitulate the synchrony seen between cNE neurons should not be surprising. The premise of the cNE algorithm (*Lopes-dos-Santos et al., 2013*) is to detect groups of neurons with reliable synchronous activity and identify these synchronous events, which are likely driven by higher-order correlations (*Montani et al., 2009*; *Ohiorhenuan et al., 2010*; *Yu et al., 2011*). These higher-order correlations could be a result of common synaptic input (*Shadlen and Newsome, 1998*), often shared by neighboring cortical neurons (*Song et al., 2005*; *Yoshimura et al., 2005*), or a result of global fluctuations driven by top-down inputs (*Goris et al., 2014*; *Schölvinck et al., 2015*). cNEs, and the statistics that we have used to describe them, are hence a useful tool to study higher-order correlations and the biological phenomena that drive them.

## cNEs are functional units that reliably represent and transmit information

Our data suggest that cNEs may be conveyors of specific environmental information that is either externally triggered or internally engaged. Acoustically driven cNEs conveyed more information in their response events than individual neurons or randomly selected groups of neurons. Since information necessarily includes timing information, this implies that cNEs are tightly locked to stimulus features that trigger ensemble responses. Additionally, since information increases as response precision increases (*Figure 9* and *Figure 9—figure supplement 1*), our results show that cNEs are more temporally reliable.

One way to examine a subset of stimulus features that trigger ensemble responses was to compare cNE STRFs against neuronal STRFs. To that end, we showed that cNE STRFs had enhanced excitatory and inhibitory subfields when compared to those of member neurons or randomly selected groups of neurons (*Figure 10* and *Figure 10—figure supplement 1*). This phenomenon could be explained by a reduction in the number of events during periods where the stimulus and the STRFs are relatively uncorrelated, or by an increase in the number of events that respond to stimulus segments with high projection values. We propose that both ideas are likely and contribute to the observed phenomena. cNE events only require a subset of member neurons to fire together (*Figure 4D and E*). Furthermore, from one cNE event to another, the neurons that contribute to any event can vary among the cNE members. Given the probabilistic nature of neuronal spiking even for the highest projection values, it suggests that cNEs integrate complementary spiking information from member neurons during stimulus presentations to increase the number of events that respond to stimulus segments with high projection values and to better encode the stimulus. On the other hand, spikes that occur during stimulus segments with low projection values do not tend to be synchronous with spikes from other neurons and are less likely to be associated with cNE events (see *Figures 4D, E*, *9A and B*).

Altogether, cNEs appear to more reliably encode features of auditory stimuli. If one purpose of synchronous firing by groups of neurons is to produce a more efficacious downstream effect than can be accomplished by a single neuron (*Buzsáki, 2010*), then cNEs may be the functional units that serve to maintain and/or enhance the fidelity of the encoding of relevant sound features. They are likely part of larger networks that can span more than a column and extend over several hierarchical levels. The relative constancy of cNEs can provide stable spatio-temporal firing patterns with transient multi-neuronal synchrony that may, among other functions, signal salient spectro-temporal events (*Hopfield and Brody, 2001*) or contribute to the learning and extraction of stimulus categorization (*Higgins et al., 2017*).

## Conclusion

In recent years, studies have identified neuronal ensembles or cell assemblies based on sophisticated statistical analyses and have ascribed functions to them based on observations of the behaviors of the identified constructs (*Bathellier et al., 2012*; *Ikegaya et al., 2004*). We extend these results to columnar AI responses by identifying cNEs and rigorously examining alternative hypotheses for their generation. Synchrony driven by the overlap of STRFs appears to be insufficient to account for the synchrony observed between neurons in the same cNE. Since we have established that AI cNEs are also found in spontaneous activity and cannot be reduced to stimulus synchronization, further

elucidating the formation of cNEs and their roles in auditory information processing and transmission are necessary next steps.

## Materials and methods

### Electrophysiology

All experiments were approved by the Institutional Animal Care and Use Committee at the University of California, San Francisco. Female Sprague-Dawley rats (200–300 g, 2–3 months old; RRID: MGI:5651135) were anesthetized with a mixture of ketamine and xylazine, supplemented with dexamethasone, atropine and meloxicam. A tracheotomy was performed to stabilize the rat's breathing. Lidocaine was injected at the sites of the tracheotomy and craniotomy before each incision. The skin, bone and dura over the right auditory cortex were removed and the brain was kept moist with silicone oil. A cisternal drain at the cisterna magna was performed to keep the brain from swelling. The primary auditory cortex (AI) was located by using multiunit responses to pure tones of different frequencies (1–40 kHz) and intensities (10–80 dB). Regions with short latencies (10–30 ms) and a tonotopic gradient in the rostrocaudal axis were identified to be AI (*Polley et al., 2007*). Recordings for all data except *Figure 8* were made using a 32-channel probe (a1 × 32-poly3, NeuroNexus, *Figure 1A*), and recording data for *Figure 8* was made using a 64-channel probe (H3, Cambridge NeuroTech), inserted perpendicular to the cortical surface to a depth of approximately 800 or 1400 µm respectively using a microdrive (David Kopf Instruments). Neural traces were band-pass filtered between 500 and 6000 Hz and were recorded to disk at 20 kHz sampling rate with an Intan RHD2132 Amplifier system. The 32-channel traces were spike-sorted offline with a Bayesian spike-sorting algorithm that included a spike-waveform decomposition procedure to extract single-unit spike timing (*Lewicki, 1994*). Neurons recorded on neighboring electrode contacts were identified via cross-correlation analysis. The 64-channel traces were spike-sorted offline using MountainSort, a fully automated spike sorter (*Chung et al., 2017*).

### Stimulus

The stimulus was either a dynamic moving ripple or a ripple noise (*Atencio et al., 2008*; *Escabi and Schreiner, 2002*). The dynamic moving ripple (DMR) was a temporally varying broadband sound (500 Hz – 40 kHz) made up of approximately 50 sinusoidal carriers per octave, each with randomized phase. The maximum spectral modulation frequency of the DMR was four cycles/oct, and the maximum temporal modulation frequency was 40 cycles/s. The maximum modulation depth of the spectro-temporal envelope was 40 dB. Mean intensity was set at 30–50 dB above the average pure tone threshold within a penetration. The ripple noise (RN) stimulus was the sum of 16 independently created DMRs. Both DMR and RN were presented as long and continuous 10-min stimuli. Spike-triggered averaging of the DMR and RN was used to recover the spectro-temporal receptive field (STRF) of each neuron (*Figure 1C*). 5-s identical segments of either stimulus were also repeatedly presented 50 times, with 0.5 s of silence between each repetition (*Escabi and Schreiner, 2002*).

### Detection of coordinated neuronal ensembles (cNEs)

All data analysis was performed in MATLAB (Mathworks). The cNE detection algorithm used in this study was based on dimensionality reduction techniques (*Lopes-dos-Santos et al., 2013*). Single-unit spike times were binned into 10-ms time bins and normalized via z-scoring. To determine the number of cNEs in each dataset, principal component analysis (PCA) was applied to the z-scored spike matrix to obtain the eigenvalue spectrum. Eigenvalues that exceeded the upper bounds of the Marčenko-Pastur distribution (*Marčenko and Pastur, 1967*) were deemed significant and represented the number of detected cNEs. The eigenvectors corresponding to significant eigenvalues were then processed using the fast independent component analysis (fastICA) algorithm, which approximates the distance to Gaussianity based on negentropy (*Hyvärinen and Oja, 1997*). The resulting independent components (ICs) represent the contribution of each neuron to each cNE. The activity of each cNE was calculated by then projecting the ICs back onto the z-scored spike matrix (*Lopes-dos-Santos et al., 2013*). To validate cNE membership and threshold cNE activity, the rows of the binned spike matrix were circularly shuffled independently. PCA/ICA was then applied to the

resulting shuffled spike matrices. Because correlations between neurons have been broken up by shuffling, no eigenvalues crossed the upper bounds of the Marčenko-Pastur distribution. However, the N-largest eigenvalues, where N was the number of significant eigenvalues derived from the original spike matrix, were processed using ICA. This process was iterated 100 times to get a normal distribution of IC weights for the shuffled condition. The upper and lower thresholds for neuronal membership to cNEs in a particular recording was set as ±1.5 standard deviations from the mean of this distribution. To threshold cNE activity, the ICs that were calculated from the original spike matrix were projected onto the shuffled spike matrices to obtain a normal distribution of cNE activity values. The threshold for considering a cNE to be active was defined by the 99.9[th] percentile of that distribution.

## STRF mutual information (MI)

The estimation of the MI between the STRF and single spikes or events was based on previous studies (*Atencio et al., 2008*; *Atencio and Schreiner, 2013*). Each stimulus segment, *s*, that preceded a cNE event or neuronal spike was projected onto the STRF via the dot product z = s • STRF. These projections were binned to form the probability distribution P(z|event). We then formed the prior probability distribution, P(z), by projecting all stimulus segments onto the STRF, regardless of event/spike occurrence. Both probability distributions were then normalized to the mean, μ, and standard deviation, σ, by x = (z - μ) / σ, to obtain P(x) and P(x|event). The MI between projections onto cNE STRFs and single events/spikes was then computed by:

$$I = \int dx P(x|event) log_2 \frac{P(x|event)}{P(x)}$$

Each information value was calculated using different fractions of the dataset for each spike train. To accomplish this, the information values were calculated over 90, 95, 97.5, 99 and 100% of the number of cNE events. This random sampling was iterated 20 times for each percentage. The means of the information calculated from the iterations of these percentages were plotted against the inverse of the percentages (1/90, 1/95, etc.). The information values were then extrapolated to infinite dataset size by fitting a line to the data and taking the ordinate intersect as the information value for unlimited dataset size. This method was used to determine the optimal time bin size for cNEs (*Figure 5A*) and to compare the STRF MI between cNEs, neurons and randomly selected subsets of neurons (termed 'random groups'; *Figure 10* and *Figure 10—figure supplement 1*).

To determine the optimal time bin size for cNEs, we binned spike trains at multiple bin sizes (2, 5, 8, 10, 15, 20, 35, 50 ms; *Figure 5*), and then applied the cNE detection algorithm. To allow comparison between cNEs, we interpolated the estimated activity vectors to obtain 1-ms resolution cNE activity vectors. For each of these vectors, we set a threshold such that each cNE had 5000 events. We then calculated the STRF for each cNE via event-triggered averaging, and the STRF MI for each bin size (*Figure 5A*).

## MI analysis on 5-s repeats

The estimation of MI between the patterns of firing over repeated trials of the same stimulus and the average firing rate of a neuron or group of neurons was based on an earlier study (*Brenner et al., 2000*). The spike or event trains obtained from single neurons, cNEs or random groups were represented in a matrix where each row represented one of the 50 trials, and each column represented a 10-ms time bin at the same point in the stimulus in each of the 50 trials. The post-stimulus time histogram (PSTH) vector, or r(t), was computed by taking the average number of spikes across each column of the matrix. The vector, r(t), was averaged to get a scalar, $\bar{r}$, or the average firing rate of each neuron, cNE or random group. The MI between the patterns of firing and the average firing rate was then computed by:

$$I = \frac{1}{T} \int_0^T dt \left( \frac{r(t)}{\bar{r}} \right) log_2 \left( \frac{r(t)}{\bar{r}} \right)$$

Each information value was calculated using different fractions of the dataset for each spike raster. To accomplish this, the information values were calculated over 90, 95, 97.5, 99 and 100% of the number of time bins of r(t). This random sampling was iterated 20 times for each percentage. The

means of the information calculated from the iterations of these percentages were plotted against the inverse of the percentages (1/90, 1/95, etc.). The information values were then extrapolated to infinite dataset size by fitting a line to the data and taking the ordinate intersect as the information for unlimited dataset size. This method was used to compare the MI provided by the rasters of responses to 5-s repeats of DMR or RN stimuli between cNEs, neurons and randomly selected sub-sets of neurons (termed 'random groups'; *Figure 9* and *Figure 9—figure supplement 1*).

## MI comparisons between cNE and neurons and cNE and random groups

For each MI comparison between a cNE and a member neuron (*Figures 9* and *10*), we sub-sampled either the neuronal spikes or the cNE events such that the number of cNE events was equal to the number of neuronal spikes.

For each MI comparison between a cNE and random groups (*Figure 9—figure supplement 1* and *Figure 10—figure supplement 1*), we had to generate random groups and the spike trains of the multi-unit activity of cNEs and random groups. To generate random groups of neurons for cNE/random-group comparisons, we selected one neuron from the cNE it was being compared to and randomly sampled from the rest of the neurons within a penetration, while excluding other member neurons of the cNE. Random groups of neurons that had more than one neuron from one of the other cNEs in the same recording were excluded as well. If 500 unique random groups of neurons could not be obtained (depending on the number of recorded neurons and the number of cNEs), this procedure was repeated to include two neurons from the cNE it was being compared to and the threshold for neurons from other cNEs in the same recording was similarly increased to two. This procedure was repeated until we obtained 500 unique random groups of neurons for each random group-cNE comparison that had as few neurons from within the same cNE as possible.

To generate rasters or spike trains, we summed the spikes of all member neurons of cNEs and neurons of random groups. For each cNE comparison to 500 iterations of random groups, we sub-sampled the number of spikes in the cNE and each of the 500 random groups to the number of spikes of the cNE or random group with the fewest spikes. Any set of cNE and random groups with fewer than 200 spikes after sub-sampling were excluded from the MI analysis to limit spurious comparisons.

## Constructing repeat ('surrogate') spike matrices and 'real' spike matrices

To verify that the synchrony observed between neurons in the same cNE cannot be explained via overlapping receptive fields or stimulus synchronization, we compared synchrony between neuronal spike trains from the same presentation of the stimulus ('real') against neuronal spike trains across different presentations of the same stimulus ('surrogate'). A brief overview of the process of constructing 'real' and 'surrogate' spike matrices is depicted in *Figure 11—figure supplement 1*. From the 24 units that had reliable firing rates across all 15 repetitions of the 10-min RN stimulus, we randomly selected 100 unique 15-neuron combinations. For each combination, we constructed 15 'real' and 'surrogate' spike matrices. Each 'real' spike matrix was constructed by taking the spike trains of the neurons in each combination from one presentation of the RN stimulus. Since there were 15 presentations of the stimulus, there were 15 'real' spike matrices for each unique 15-neuron combination. Each 'surrogate' spike matrix was constructed by first randomly sorting each 15-neuron combination. The spike train for the first neuron in the combination was then taken from the first presentation of the stimulus, the second neuron from the second presentation, etc., to get one 'surrogate' spike matrix. This was repeated 14 additional times, each time by circularly shifting the randomly sorted 15-neuron combination vector by 1, to get 15 different 'surrogate' spike matrices for each 15-neuron combination. We then calculated the number of cNEs detected and mean cNE size for each of the 'real' and 'surrogate' spike matrices. Finally, by averaging across the 15 spike matrices for each measure (number of cNEs or mean cNE size), label ('real' or' surrogate') and unique combination, we obtained 100 data points for each measure and label. The results are plotted in *Figure 11A and B*.

## Simulated spike trains

We used three simulated spike trains in this study to validate our observations, the 'shuffled', the 'preserved receptive field' ('PR'), and the 'dichotomized Gaussian' ('DG') models. The 'shuffled' model was computed by circularly shifting the rows of the spike matrix (where each row represents the binned spike train of one neuron) by a random number of bins, independently of other rows.

The 'PR' model was simulated based on each neuron's STRF. First, each stimulus segment was correlated with the STRF to obtain P(z), the distribution of projection values without regard to a spike. Next, the full range of P(z) was divided into 15 equal-sized projection value bins. Projection values in each bin represent stimulus segments that were similarly correlated with the STRF. Projection value bins contained multiple projection values, with each of them corresponding to a different time within the stimulus. To obtain the distribution of projection values that corresponded to a spike for each projection value bin, the number of projection values corresponding to spikes was divided by the total number of projections values for that bin regardless of whether a spike occurred. This normalized distribution was P(spk|z). To shuffle spike times with respect to P(spk|z), spikes corresponding to a specific projection value were randomly assigned to different time bins that corresponded to projection values in the same projection value bin. This shuffling procedure preserves the distribution of projection values while creating a random spike train (*Figure 11—figure supplement 2*).

The 'DG' model simulates spike trains that preserve the experimentally derived mean firing rate for each neuron as well as the PWC between each pair of neurons (*Macke et al., 2011*; *2009*). Implementation of the 'DG' model with MATLAB can be found in *Macke et al. (2009)* and is available at https://github.com/mackelab/CorBinian (*Macke, 2017*). The 'DG' model requires and produces binary spike trains and works best if the temporal resolution is high enough so that a maximum of one spike occurs in each time bin. However, since we also wanted to capture the temporal correlations between neurons, we simulated 'DG' spike matrices at 5-ms time resolution. We then binned the spike trains at a temporal resolution of 10 ms and completed the same analyses that were used to process the experimental data.

We iterated each of the 16 recordings 200 times per model and calculated the mean of cNE number and size for each model and recording. These statistics were compared with those of the real spike trains in *Figure 11C and D*.

## Calculation of coincidence ratios (CRs)

The CR was defined as the ratio of the number of time bins in which all neurons in the pre-defined group spike together, to the number of time bins in which the neuron with the lowest firing rate is active. An illustration of this calculation is shown in *Figure 11—figure supplement 4A*. 400 unique combinations of 2, 3, 4 or 5 neurons within the same cNE were randomly selected. For each combination of neurons, we obtained one 'real' and one 'repeat' CR values. 'Real' CRs were calculated by computing the CR between neuronal spike trains from one presentation of the stimulus. 'Repeat' CRs were calculated by computing the CR between neuronal spike trains from different presentations of the stimulus. Each combination of neurons also had 1000 'shuffle', 'PR' and 'DG' CR values. These were computed by iterating the shuffle and PR models 1000 times and calculating the CR for each iteration of each model. The median CR value for each model was calculated and compared to the single 'real' and 'repeat' CR values. An example of this comparison for a 2-neuron combination is shown in *Figure 11—figure supplement 4B*. CR values for 2- and 3-neuron combinations are plotted in *Figure 11E and F*. Each set of CR values is normalized to the 'real' CR value to determine how much of the real coincidence can be approximated by each data manipulation (*Figure 11E and F*).

## Acknowledgements

The authors would like to thank Michael Brainard and Massimo Scanziani for invaluable scientific discussions; Brian Malone, Tatyana Sharpee and Brett Mensh for feedback on the manuscript; Vitor Lopes-dos-Santos for providing his dimensionality-reduction MATLAB code; and Jakob Macke and Marcel Nonnenmacher for providing their dichotomized Gaussian MATLAB code and guidance on its application.

## Additional information

### Funding

| Funder | Grant reference number | Author |
|---|---|---|
| National Institute on Deafness and Other Communication Disorders | DC02260 | Craig A Atencio Christoph E Schreiner |
| Coleman Memorial Fund | | Craig A Atencio Christoph E Schreiner |
| Hearing Research Incorporate, San Francisco | | Craig A Atencio Christoph E Schreiner |
| Agency for Science, Technology and Research, Singapore | National Science Scholarship | Jermyn Z See |

The funders had no role in study design, data collection and interpretation, or the decision to submit the work for publication.

### Author contributions

Jermyn Z See, Conceptualization, Data curation, Formal analysis, Validation, Investigation, Visualization, Methodology, Writing—original draft, Writing—review and editing; Craig A Atencio, Conceptualization, Formal analysis, Investigation, Methodology, Writing—review and editing; Vikaas S Sohal, Supervision, Writing—review and editing; Christoph E Schreiner, Conceptualization, Resources, Formal analysis, Supervision, Funding acquisition, Validation, Investigation, Methodology, Project administration, Writing—review and editing

### Author ORCIDs

Jermyn Z See http://orcid.org/0000-0002-8372-0753
Vikaas S Sohal http://orcid.org/0000-0002-2238-4186
Christoph E Schreiner http://orcid.org/0000-0002-4571-4328

### Ethics

Animal experimentation: This study was performed in strict accordance with the recommendations in the Guide for the Care and Use of Laboratory Animals of the National Institutes of Health. All of the animals were handled according to approved institutional animal care and use committee (IACUC) protocols (#100-17) of the University of California, San Francisco. The protocol was approved by the IACUC of the University of California, San Francisco (Protocol Number: AN165706-02). All surgery was performed under ketamine/xylazine anesthesia, and every effort was made to minimize suffering.

### Decision letter and Author response

Decision letter https://doi.org/10.7554/eLife.35587.030
Author response https://doi.org/10.7554/eLife.35587.031

## Additional files

### Supplementary files

• Supplementary File 1. Details of statistical tests used in the study, including the type of statistical tests used, the sample sizes (N), and the exact p-values.
DOI: https://doi.org/10.7554/eLife.35587.025

• Transparent reporting form
DOI: https://doi.org/10.7554/eLife.35587.026

## Data availability

Single-unit extracellular electrophysiological data have been deposited in CRCNS.org under DOI citation http://dx.doi.org/10.6080/K09021X1

The following dataset was generated:

| Author(s) | Year | Dataset title | Dataset URL | Database, license, and accessibility information |
|---|---|---|---|---|
| See JZ, Atencio CA, Schreiner, CE | 2018 | High-density extracellular recordings from the primary auditory cortex in anesthetized rats listening to dynamic broadband stimuli. | http://dx.doi.org/10.6080/K09021X1 | Publicly available at the Collaborative Research in Computational Neuroscience data sharing website (http://crcns.org/) |

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
