## [Decision Letter]

[Editors’ note: a previous version of this study was rejected after peer review, but the authors submitted for reconsideration. The first decision letter after peer review is shown below.]

Thank you for submitting your work entitled "Coordinated Neuronal Ensembles in Primary Auditory Cortical Columns" for consideration by *eLife*. Your article has been reviewed by three peer reviewers, and the evaluation has been overseen by a Reviewing Editor and a Senior Editor. The following individual involved in review of your submission has agreed to reveal their identity: Shihab Shamma (Reviewer #1).

Our decision has been reached after consultation between the reviewers. Based on these discussions and the individual reviews below, we regret to inform you that your work will not be considered further for publication in *eLife*.

All reviewers agree that the paper focuses on an important question, which is how ensembles of synchronized neurons in cortex function and represent stimulus information. Nevertheless, and as set out in the appended reviews, they felt that key elements of this claim were insufficiently supported. For example, with respect to the methods, the reviewers felt it was not clear what assumptions were made, and how the results were affected by grouping criteria. There were also concerns about the criteria for establishing whether or not receptive fields overlapped. This is important for ruling out the possibility that neurons responded to a similar feature of stimulus nonlinearly. The reviewers also felt that the functional consequences of these data remain unclear.

Reviewer #1:

This is a well-written paper that provides solid analysis of an exceptional set of data from primary auditory cortex. On the positive side, the paper advances the concept of cNE in auditory cortex and offers a suite of algorithms and tests to define and identify them. Much of the paper is a solid analysis that deals convincingly with the many possible confounds. For the most part, what is discussed and demonstrated is well-presented, and thus I have little to complain about. All in all, I found the paper enjoyable and informative. What I find less satisfying is what is *not* in the manuscript, as I will explain:

I found the following sentence very interesting, but is unfortunately undeveloped, and is not addressed in the paper at all:

"If one goal of synchronous firing by groups of neurons is to produce a more efficacious downstream effect than can be accomplished by a single neuron (Buzsáki, 2010), then cNEs may be the functional units that serve to maintain and/or enhance the fidelity of the encoding of relevant sound features".

So what are examples of these features? In other words, the functional significance of these cNE is not addressed at all in this paper except by the general statements in the end. The experiments exploited a broadband complex stimulus, and even showing the STA's measured from the firings of the cNE's would have provided (I believe) really interesting insights into what stimulus features these cNEs are trying to enhance. Such measurements would justify and use the analyses shown, and elevate the paper from a "Methods" paper to something much more, which it deserves to be with a little more analysis of the cNE data. Obviously, many future experiments might engage animals in behaviors or go after the features in speech and music and so on, or to find out if these cNE's remain stable or not, and whether they are somehow related to "Brain States".

Reviewer #2:

The paper "Coordinated Neuronal Ensembles in Primary Auditory Cortical Columns" uses multi electrode recordings to study the synchronized activity of groups of neurons and investigate enhanced information processing by their coordinated activity. The question at the center of this paper is very important, however the paper doesn't really deliver what is promised. The shortcomings of the study are discussed in detail below.

To begin with, how exactly the ensemble of synchronized neurons are identified is unclear. It is mentioned that they apply independent component analysis to the most significant PCA coefficients, but no more detail is provided. While the method is demonstrated using a simulated example, it is unclear what assumptions are being made, what are the limitations of this method, and how the results discussed are affected by this grouping criteria. In short, identifications of neural ensemble in this paper is treated as a trivial, solved problem, which is not the case because the choice of the method can hinder the interpretation of the results.

As an example, the authors show that PWC between CNE members vary from non-members. However, given that the CNEs were chosen based on correlation patterns (PCA/ICA of autocorrelation matrix), this observation is of course expected from the selection criteria.

Another major claim of the study is enhanced information transfer. However, the measure of information used in this study does not really reflect whether CNE carries more information about the stim, the state, or about any other factor as claimed. The measure used is basically an entropy of PSTH, measuring how far from uniform the average spike train is. In other word, it is calculated by averaging the neural responses over 50 repetitions of the same stimulus, and then the entropy of this average is computed. So, this measure only reflects how "bumpy" the average response is. Again, the finding of higher entropy for CNE neurons is expected from the selection criteria of these neurons. If the averaging is done over uncorrelated neurons, the average will be more flat. If they are from a CNE, they by selection have higher correlation and as a result their average will be less uniform (average is indeed what the first PC of the data approximates). Therefore, this finding is not really compelling and informative. I suggest methods such as decoding or "Mutual" information instead of plain entropy. In particular, answering "what" information is more reliably represented and "how" is crucial for the claims of the study.

Moreover, the method used identifies neurons that are co-activated. But obviously if they are fully correlated, then there will be no added information by the ensemble. This issue is ignored, and the authors have only focused on similarity of CNE responses, and not the complementary components encoded by ensemble members.

Finally, the evidence for the claim that the CNEs are not based on overlapping receptive fields is rather weak. The evidence shown is based on STRF, which is a lousy model of neuron's actual receptive field. Therefore, they cannot rule out the possibility that neurons responded to a similar feature of stimulus nonlinearly.

From Discussion "Instead, cNE activity represents important stimulus information embedded in broader network-related contexts". This is indeed a very interesting claim, but I do not believe the author have shown compelling evidence for it.

Reviewer #3:

This is a thorough study identifying cell assemblies in auditory cortex. Spiking events from these cell assemblies carried more information than spikes from single neurons or randomly selected groups of neurons with the same spike/event rate. The authors use a number of controls to demonstrate that the identified cell assemblies could not be solely accounted for by the receptive field similarity.

The analysis in the subsection “Identification of coordinated neuronal ensembles (cNEs)” correlating results obtained full datasets and the first three quarters is somewhat confusing. Some of the overlap in the results will be due to overlap in the data. One possibility is to use bootstrap correction to report the expected correlation that has been compensated for the dataset overlap. A more standard alternative would be to compare cell assemblies identified from different 3/4 of the datasets, and then if they perfectly overlap that would be one answer. If the overlap is less than 100% this can again be corrected using bootstrap formula from the Efron and Tibshirani book "An introduction to the bootstrap".

[Editors’ note: what now follows is the decision letter after the authors submitted for further consideration.]

Thank you for submitting your article "Coordinated Neuronal Ensembles in Primary Auditory Cortical Columns" for consideration by *eLife*. Your article has been reviewed by three peer reviewers, and the evaluation has been overseen by a Reviewing Editor and Andrew King as the Senior Editor. The reviewers have opted to remain anonymous.

The reviewers have discussed the reviews with one another and the Reviewing Editor has drafted this decision to help you prepare a revised submission.

We suggest that you make the analyses of the noise correlations more explicit, and test whether the synchrony you observe goes beyond that expected for pairs. To do this, you could compare against surrogate data sampled from a model, e.g. Ising or dichotomized Gaussian, with the same pairwise correlations but no higher-order correlations.

[Editors' note: further revisions were requested prior to acceptance, as described below.]

Thank you for sending your revised article entitled "Coordinated Neuronal Ensembles in Primary Auditory Cortical Columns" for peer review at eLife. Your article is being evaluated by 3 peer reviewers, and the evaluation is being overseen by a Reviewing Editor and Andrew King as the Senior Editor.

Your article has been favorably reviewed, but there is one remaining reviewer comment that needs to be addressed before a final decision can be made.

Reviewer:

The use of the DG model is appropriate, but I wonder why the authors used a model that matches the pairwise correlations only at zero delay? They point out that this produces simulated spike trains that differ from the real data in two ways: 1) higher-order correlations at all delays and 2) second-order correlations at non-zero delays. If the goal is to quantify the impact of higher-order correlations, this might not the best way to do it. The DG framework can be used to generate spike trains that match the second-order correlations at all delays. Wouldn't that be better for this purpose?

---

## [Author Response]

[Editors’ note: the author responses to the first round of peer review follow.]

Reviewer #1:This is a well-written paper that provides solid analysis of an exceptional set of data from primary auditory cortex. On the positive side, the paper advances the concept of cNE in auditory cortex and offers a suite of algorithms and tests to define and identify them. Much of the paper is a solid analysis that deals convincingly with the many possible confounds. For the most part, what is discussed and demonstrated is well-presented, and thus I have little to complain about. All in all, I found the paper enjoyable and informative. What I find less satisfying is what is not in the manuscript, as I will explain:I found the following sentence very interesting, but is unfortunately undeveloped, and is not addressed in the paper at all:"If one goal of synchronous firing by groups of neurons is to produce a more efficacious downstream effect than can be accomplished by a single neuron (Buzsáki, 2010), then cNEs may be the functional units that serve to maintain and/or enhance the fidelity of the encoding of relevant sound features".

To assess this more specific idea of coordinated activity in the form and function of ‘assemblies’ requires that source and target activity be monitored. Activity from within a cortical column may contain instances of such sequential activation given that we explicitly included that possibility with our selection of the 10 ms synchrony window. Actually, the few cNEs that extended across several layers (Figure 7F) may be examples of such ‘projection assemblies’. However, to separate sources and targets within a column requires much more data than available. To resolve this issue, measurements are needed that include more spatially segregated sites of sources and targets, e.g., from thalamus, IC, or other cortical fields.

So what are examples of these features? In other words, the functional significance of these cNE is not addressed at all in this paper except by the general statements in the end. The experiments exploited a broadband complex stimulus, and even showing the STA's measured from the firings of the cNE's would have provided (I believe) really interesting insights into what stimulus features these cNEs are trying to enhance. Such measurements would justify and use the analyses shown, and elevate the paper from a "Methods" paper to something much more, which it deserves to be with a little more analysis of the cNE data. Obviously, many future experiments might engage animals in behaviors or go after the features in speech and music and so on, or to find out if these cNE's remain stable or not, and whether they are somehow related to "Brain States".

We agree and have added a figure to give an initial idea of the features that are enhanced (Figure 10). The implications of this result are further discussed in the Discussion section (subsection “cNEs are functional units that reliably represent and transmit information”, second paragraph) and a much more detailed receptive field analysis will be done in a future manuscript.

Reviewer #2:The paper "Coordinated Neuronal Ensembles in Primary Auditory Cortical Columns" uses multi electrode recordings to study the synchronized activity of groups of neurons and investigate enhanced information processing by their coordinated activity. The question at the center of this paper is very important, however the paper doesn't really deliver what is promised. The shortcomings of the study are discussed in detail below.To begin with, how exactly the ensemble of synchronized neurons are identified is unclear. It is mentioned that they apply independent component analysis to the most significant PCA coefficients, but no more detail is provided. While the method is demonstrated using a simulated example, it is unclear what assumptions are being made, what are the limitations of this method, and how the results discussed are affected by this grouping criteria. In short, identifications of neural ensemble in this paper is treated as a trivial, solved problem, which is not the case because the choice of the method can hinder the interpretation of the results.

We have added additional information about the method in the Results (subsection “Identification of coordinated neuronal ensembles (cNEs)”, first paragraph) to discuss this.

As an example, the authors show that PWC between CNE members vary from non-members. However, given that the CNEs were chosen based on correlation patterns (PCA/ICA of autocorrelation matrix), this observation is of course expected from the selection criteria.

It is indeed unsurprising that PWCs between cNE members are sharper than PWCs between non-cNE members, given how we determine cNEs and the high correlation between Pearson’s correlation coefficient and sharpness of PWCs (Figure 2B). It was not our intention to tout that as the main point of Figure 6, and have changed the heading of that section to reflect that (“Synchrony between cNE members cannot be fully explained by receptive field overlap”). Rather it illustrates that the correlation sharpness for cNE members is narrower than both the bin size constraining spiking synchrony and the receptive field (RF) correlation width.

Another major claim of the study is enhanced information transfer. However, the measure of information used in this study does not really reflect whether CNE carries more information about the stim, the state, or about any other factor as claimed. The measure used is basically an entropy of PSTH, measuring how far from uniform the average spike train is. In other word, it is calculated by averaging the neural responses over 50 repetitions of the same stimulus, and then the entropy of this average is computed. So, this measure only reflects how "bumpy" the average response is. Again, the finding of higher entropy for CNE neurons is expected from the selection criteria of these neurons. If the averaging is done over uncorrelated neurons, the average will be more flat. If they are from a CNE, they by selection have higher correlation and as a result their average will be less uniform (average is indeed what the first PC of the data approximates). Therefore, this finding is not really compelling and informative. I suggest methods such as decoding or "Mutual" information instead of plain entropy. In particular, answering "what" information is more reliably represented and "how" is crucial for the claims of the study.

The reviewer is correct in that the entropy due to the inherent noisiness of the responses (noise entropy) must be balanced against the entropy due to the signal to calculate mutual information. For our analysis, we applied the approach of Brenner et al. (2000), and adopted their metric for mutual information. In their paper, they do indeed derive this mutual information metric from the difference between two entropies (see pages 1534-1539 of their paper for mathematical derivation). Thus, our approach does appropriately capture stimulus information, and is not equivalent to simply measuring the entropy of PSTHs.

The finding of higher entropy for cNE neurons is not necessarily expected. The detection of cNEs only requires neurons to be coincident in 10-ms time bins; this does not require for the neurons to be coincident in 10-ms time bins and be responsive to specific characteristics in the stimulus. In fact, we have cNEs that are not very stimulus-responsive and their correlated firings are not reliable with respect to the stimulus. These cNEs end up having flat PSTHs as well, and random groups of neurons that are all stimulus responsive (and hence can be synchronized by the stimulus) can end up having “peakier” PSTHs (and hence more information) than the cNEs that are not stimulus-responsive.

We have added another mutual information analysis that takes into account the receptive fields of neurons and cNEs (Figure 10). This is meant to answer “what” information is more reliably represented. “How” it is represented is discussed in the Discussion (subsection “cNEs are functional units that reliably represent and transmit information”, second paragraph) and further analyses will be done in another manuscript, given the substantial length of that analysis.

Moreover, the method used identifies neurons that are co-activated. But obviously if they are fully correlated, then there will be no added information by the ensemble. This issue is ignored, and the authors have only focused on similarity of CNE responses, and not the complementary components encoded by ensemble members.

The degree of co-activation is, obviously, quite low, given average correlation coefficients of < 0.08. Nonetheless, the analysis of the complementary components encoded by ensemble members and the whole cNE is highly relevant and will be treated in detail elsewhere given its extensive nature. We have added a section in the Discussion that discusses this idea (subsection “cNEs are functional units that reliably represent and transmit information”, second paragraph).

Finally, the evidence for the claim that the CNEs are not based on overlapping receptive fields is rather weak. The evidence shown is based on STRF, which is a lousy model of neuron's actual receptive field. Therefore, they cannot rule out the possibility that neurons responded to a similar feature of stimulus nonlinearly.

There are several independent lines of reasoning to address this point.

i) The linear STRF is, of course, not a complete representation of the stimulus-preference of a neuron. Additional stimulus features may activate neurons, often in nonlinear fashion, as has been demonstrated using multi-filter approaches (e.g., Atencio et al., 2008; Harper et al., 2016). Nonetheless, the main (linear) filters appear to carry the most information. The similarity of only the main features captured by these first filters (determined by STAs) are used here, and cannot exclude the possibility of ‘hidden’ feature-driven aspects of the cNE neurons. Preliminary analysis of second-order filters, however, show an even lower degree of similarity between cNE members and non-members. A more detailed analysis of these aspects will be forthcoming.

ii) Comparison of responses to repeated stimuli (Figures 11A and 11B) reflects a model-free (non-STRF based) analysis that basically ruled out that neurons consistently responded to linearly or nonlinearly encoded similar stimulus features.

iii) The new spontaneous-evoked cNE comparison (Figure 8) further supports the idea that the activation of these cNEs are not driven by receptive field similarities alone.

From Discussion "Instead, cNE activity represents important stimulus information embedded in broader network-related contexts". This is indeed a very interesting claim, but I do not believe the author have shown compelling evidence for it.

We have now included Figure 10 to illustrate that cNE STRFs have enhanced excitatory and inhibitory subfields when compared against single neurons STRFs that shows evidence for the above claim.

Reviewer #3:[…] The analysis in the subsection “Identification of coordinated neuronal ensembles (cNEs)” correlating results obtained full datasets and the first three quarters is somewhat confusing. Some of the overlap in the results will be due to overlap in the data. One possibility is to use bootstrap correction to report the expected correlation that has been compensated for the dataset overlap. A more standard alternative would be to compare cell assemblies identified from different 3/4 of the datasets, and then if they perfectly overlap that would be one answer. If the overlap is less than 100% this can again be corrected using bootstrap formula from the Efron and Tibshirani book "An introduction to the bootstrap".

We have corrected this inaccuracy and now use the first half of the stimulus (training) to predict the cNE activity of the second half of the stimulus (test) avoiding overlap between the training and test datasets (Figure 4—figure supplement 3). The findings and their interpretation are not affected.

[Editors' note: the author responses to the re-review follow.]

We suggest that you make the analyses of the noise correlations more explicit.

We implemented this suggestion and completed an explicit noise correlation assessment by analyzing the responses of pairs of neurons to 5-second repeated stimuli. We report these results in Figure 9—figure supplement 2. Our analysis showed that pairs of neurons from the same cNE had significantly higher noise correlations than pairs of neurons not within the same cNE.

And test whether the synchrony you observe goes beyond that expected for pairs. To do this, you could compare against surrogate data sampled from a model, e.g. Ising or dichotomized Gaussian, with the same pairwise correlations but no higher-order correlations.

We have followed this suggestion and compared the experimental data to surrogate data derived from a ‘dichotomized Gaussian’ (‘DG’) model. We used the open-source library available at https://github.com/mackelab/CorBinian to implement the ‘DG’ model. The library allowed us to produce surrogate spike trains that had the same pairwise correlation coefficients as the experimental data but had no higher-order correlations. We have incorporated these results into Figures 11C–F, as well as Figure 11—figure supplements 3 and 4.

The ‘DG’ model is based on a multivariate Gaussian analysis, which generates spike trains that match the neuronal mean firing rates and the second-order correlations of the experimental spike train data. We created the model spike trains at the same native resolution as those in the experimental data, which allowed us to carry out the identical cNE analyses on the ‘DG’ surrogate data that we had completed on the experimental data.

We found that the ‘DG’ model allowed us to identify a similar number of cNEs from the surrogate spike trains as we had identified from the experimental spike trains (Figure 11C).

However, membership in a cNE is determined by local moments of multi-neuronal synchrony that occur across the entire recording period. Further, ICA is a required step beyond the pairwise correlation analysis and must be applied to assign shared neurons into appropriate cNEs. Thus, we found that ‘DG’ cNEs were smaller in size (Figure 11D) as a result of the lack of temporal correlations and higher-order, spike-time synchrony that are present in the experimental data.

The importance of synchrony was further confirmed when we found a lower coincidence ratio (CR) for pairs of 2 neurons for ‘DG’ spike trains (Figure 11E). This indicates that even for pairs of neurons, temporal correlations provide a substantial contribution to the cNE analysis. Further, this implies that merely matching the rate and pairwise-correlation statistics is not adequate to model simultaneous recorded auditory cortical columnar spike trains. Auditory cortical columnar spike trains contain strong inter-neuronal synchrony, which can only be captured with higher-order correlation analyses.

[Editors' note: further revisions were requested prior to acceptance, as described below.]Your article has been favorably reviewed, but there is one remaining reviewer comment that needs to be addressed before a final decision can be made.Reviewer:The use of the DG model is appropriate, but I wonder why the authors used a model that matches the pairwise correlations only at zero delay? They point out that this produces simulated spike trains that differ from the real data in two ways: 1) higher-order correlations at all delays and 2) second-order correlations at non-zero delays. If the goal is to quantify the impact of higher-order correlations, this might not the best way to do it. The DG framework can be used to generate spike trains that match the second-order correlations at all delays. Wouldn't that be better for this purpose?

Due to computational limitations, the most feasible remedy was to use the

basic dichotomized Gaussian (‘DG’) framework to model second-order correlations at zero delays using 5-ms time bins. Using 5-ms time bins fulfills the requirement that the spike trains be binary, and also allows us to include most of the important temporal correlations between neurons from the same cNE. Neurons within the same cNE have been shown to have very sharp pairwise-correlations with median sharpness values of approximately 2-4 ms (Figure 6G – 6J, Figure 6—figure supplement 1) and have peak delays that are 2 ms or less. Therefore, this time resolution appropriately accounts for second-order correlations at non-zero delays (Figure 11—figure supplement 3C).

We show that even though this ‘DG’ framework seems to overestimate a subset of Pearson’s correlation values between pairs of neurons, leading to a larger number of detected cNEs (Figure 11C), it does not replicate higher-order correlations that are present in the real data, since cNE sizes in the ‘DG’ spike trains are smaller than that of the real spike trains (Figure 11D).

The importance of higher-order correlations in the real spike train was further

confirmed when we found a lower coincidence ratio (CR) for pairs of 3 neurons for ‘DG’ spike trains (Figure 11F). This implies that merely matching the rate and

pairwise-correlation statistics is not adequate to model simultaneous recorded

auditory cortical columnar spike trains. Auditory cortical columnar spike trains contain strong inter-neuronal synchrony, which can only be captured with higher-order correlation analyses.